# From Scarcity to Efficiency: Preference-Guided Learning for Sparse-Reward Multi-Agent Reinforcement Learning

## Abstract

We study the problem of online multi-agent reinforcement learning (MARL) in environments with sparse rewards, where reward feedback is not provided at each interaction but only revealed at the end of a trajectory. This setting, though realistic, presents a fundamental challenge: the lack of intermediate rewards hinders standard MARL algorithms from effectively guiding policy learning. To address this issue, we propose a novel framework that integrates online inverse preference learning with multi-agent on-policy optimization into a unified architecture. At its core, our approach introduces an implicit multi-agent reward learning model, built upon a preference-based value-decomposition network, which produces both global and local reward signals. These signals are further used to construct dual advantage streams, enabling differentiated learning targets for the centralized critic and decentralized actors. In addition, we demonstrate how large language models (LLMs) can be leveraged to provide preference labels that enhance the quality of the learned reward model. Empirical evaluations on state-of-the-art benchmarks, including MAMuJoCo and SMACv2, show that our method achieves superior performance compared to existing baselines, highlighting its effectiveness in addressing sparse-reward challenges in online MARL.

## 1 Introduction

Cooperative multi-agent reinforcement learning (MARL) has emerged as a powerful paradigm for solving sequential decision-making problems in domains where multiple agents must coordinate to achieve a common objective. Important applications include autonomous driving (Shalev-Shwartz et al., 2016), robotics and swarm control (Hüttenrauch et al., 2017), network traffic management (Chu et al., 2020), and large-scale strategy games (Vinyals et al., 2019). Despite its potential, MARL remains challenging due to the non-stationarity introduced by simultaneously learning agents, the difficulty of credit assignment across agents, and the scalability issues associated with high-dimensional joint action spaces (Zhang et al., 2021).

A particularly demanding setting arises under *sparse rewards*, where agents only receive feedback at the end of a trajectory or episode, e.g., a win/loss signal in real-time strategy environments such as SMAC or SMACv2 (Samvelyan et al., 2019; Ellis et al., 2023). This setting is both common and practically relevant, as in many real-world cooperative tasks intermediate reward signals are unavailable or difficult to specify. However, sparse rewards exacerbate the intrinsic challenges of MARL: learning becomes highly sample-inefficient, exploration is significantly harder, and assigning credit to individual agent actions becomes even more ambiguous (Jaques et al., 2019; Hu et al., 2020).

While much progress has been made in RL from sparse feedback in the single-agent setting (Christiano et al., 2017; Ibarz et al., 2018; Zhang et al., 2023a), existing work on online MARL typically assumes dense and frequent reward feedback. Approaches such as value-decomposition methods (Sunehag et al., 2018; Rashid et al., 2020a) and policy-gradient variants of MAPPO (Yu et al., 2022) were not explicitly designed to cope with trajectory-level sparse rewards. As a result, there remains a gap in methods that can efficiently leverage sparse feedback for cooperative MARL.

In this paper, we address the fundamental challenge of sparse rewards in online MARL by proposing a unified and effective learning framework that integrates recent advances across several subfields, including PbRL, online MARL, and LLMs. Our contributions are summarized as follows.

**First**, our work approaches the sparse-reward MARL problem from a new perspective. Instead of relying on supervised learning to regress noisy episodic rewards—an approach often brittle and sensitive to sparse signals—we transform these rewards into trajectory preferences, a more robust and flexible form of supervision. Preferences over trajectory pairs are derived either via direct episodic reward comparison or, when interpretable features are available, with the assistance of an LLM, which enhances preference learning by incorporating both quantitative outcomes (e.g., cumulative rewards, success rates) and qualitative cues extracted from trajectories.

**Second**, we propose an integrated online centralized training decentralized execution (CTDE) paradigm that incorporates preference learning into on-policy optimization for multi-agent learning. Central to our approach is a dual-advantage value decomposition within a PPO-based approach: a global advantage for the centralized critic and local advantages for decentralized actors. These advantage estimates are derived from decomposed $Q$- and $V$-values tailored for multi-agent preference-based learning. This design enables a principled separation of global coordination from local credit assignment, yielding improved training stability and sample efficiency under sparse feedback.

**Third**, we provide comprehensive analysis demonstrating the robustness and theoretical sound of our learning framework. Specifically, we show that the learned reward converges to a behaviorally indistinguishable surrogate of the true reward, ensuring that optimal policies remain aligned with the underlying objective even in the absence of exact reward specification. Moreover, we prove that optimizing decentralized policies with respect to their local advantages is consistent with optimizing the global joint policy: the global policy gradient can be expressed as a weighted sum of local gradients. This result guarantees that decentralized updates remain aligned with the global objective.

Finally, we evaluate our method on challenging cooperative MARL benchmarks, including SMACv2 and MAMuJoCo (Ellis et al., 2023; de Witt et al., 2020). Across both domains, our approach consistently outperforms strong baselines, achieving higher final performance and superior sample efficiency in sparse-reward settings. Ablation studies further demonstrate the importance of combining local and global advantage streams, as well as the impact of different LLM-guided preference models.

## 2 RELATED WORK

**Multi-Agent Reinforcement Learning (MARL).** CTDE is the dominant framework in cooperative MARL, enabling agents to learn from global information during training while acting independently at execution (Foerster et al., 2018; Lowe et al., 2017; Rashid et al., 2020b; Sunehag et al., 2017; Kraemer & Banerjee, 2016). Within this paradigm, QMIX (Rashid et al., 2020b) introduced a mixing network and hypernetwork (Ha et al., 2017) to factorize joint value functions, laying the groundwork for more expressive methods like QTRAN Son et al. (2019) and QPLEX (Wang et al., 2020), and weighted QMIX (Rashid et al., 2020a). In parallel, policy-gradient approaches—notably extensions of PPO Schulman et al. (2017)—have been adapted for CTDE settings (Yu et al., 2022; Bui et al., 2024b; Kuba et al., 2021), offering better scalability in high-dimensional or continuous action spaces. The versatility of CTDE has also driven its adoption in adjacent areas ranging from imitation learning to preference learning in multi-agent settings (Ho & Ermon, 2016; Fu et al., 2017; Bui et al., 2024a; 2025a; Kang et al., 2024; Zhang et al., 2024; Bui et al., 2025b; Pan et al., 2022; Shao et al., 2024; Wang et al., 2022; Yang et al., 2021). Our work presents a seamless end-to-end CTDE pipeline for *sparse-reward* MARL, integrating implicit reward learning with policy optimization (i.e., a PPO-based extension) under a shared value decomposition strategy.

**Sparse Rewards in Reinforcement Learning.** A common strategy to address the sparse-reward challenge in RL is reward redistribution, which transforms delayed, episodic rewards into denser proxy signals that provide more immediate feedback throughout a trajectory. Most existing approaches to reward redistribution fall into three main categories: (i) Reward shaping (Hu et al., 2020; Tambwekar et al., 2018); (ii) Intrinsic reward design, which introduces auxiliary objectives to encourage exploration (Rajeswar et al., 2022; Pathak et al., 2017; Zheng et al., 2021; Colas et al., 2020); and (iii) Return decomposition, which breaks down cumulative rewards to assign credit more precisely across time (Arjona-Medina et al., 2019; Patil et al., 2020; Liu et al., 2019; Widrich et al., 2021; Ren et al.,

2021; Gangwani et al., 2020; Lin et al., 2024). Beyond these, recent work has explored alternative redistribution principles, such as causal credit assignment (Zhang et al., 2023a) and LLM-guided reward attribution Qu et al. (2025). While most of this research focuses on single-agent settings, there has been growing interest in MARL. A few recent studies have applied attention-based models to decompose returns across both time and agents (She et al., 2022; Xiao et al., 2022; Chen et al., 2024; Kapoor et al., 2025). In contrast, our work targets the sparse-reward MARL setting from a different angle. Rather than relying on complex attention mechanisms to estimate episodic rewards, we propose to convert sparse rewards into trajectory preferences—a less restrictive but more robust and intuitive form of feedback. We then leverage recent advances in offline preference-based RL to effectively learn policies in this online MARL setting with limited reward signals.

**Preference-Based Reinforcement Learning (PbRL).** PbRL trains policies using preference data, typically via pairwise trajectory comparisons. A common approach is to follow a two-stage pipeline: a reward function is first inferred using supervised learning (e.g., Bradley-Terry model), followed by standard RL for policy optimization (Choi et al., 2024; Christiano et al., 2017; Gao et al., 2024; Hejna III & Sadigh, 2023; Lee et al., 2021; Ibarz et al., 2018; Kim et al., 2023; Mukherjee et al., 2024; Zhang et al., 2023b). Alternatively, single-stage PbRL methods have been proposed, which learn policies directly from preferences by optimizing carefully designed objectives (An et al., 2023; Hejna et al., 2023; Kang et al., 2023; Hejna & Sadigh, 2024). While the former often suffers from high variance and instability, the latter offers improved stability and performance due to its simpler single-stage optimization. Recently, PbRL has been extended to multi-agent scenarios (Bui et al., 2025a; Kang et al., 2024; Zhang et al., 2024). Notably, all these existing methods operate in *offline* settings, where the agent learns from pre-collected data without environment interaction. In contrast, we tackle the more challenging problem of *online* MARL with sparse feedback, where instability arises from dynamic agent interactions and delayed rewards. We draw on insights from offline multi-agent PbRL to extract implicit dense rewards from sparse signals—augmented by LLM guidance when available—all within a unified CTDE framework for efficient online policy learning.

# 3 ONLINE MARL WITH SPARSE REWARDS

We focus on the setting of cooperative MARL, which can be modeled as a multi-agent POMDP, $\mathcal{M} = \langle \mathcal{S}, \mathcal{A}, P, r, \mathcal{Z}, \mathcal{O}, n, \mathcal{N}, \gamma \rangle$, where $n$ is the number of agents and $\mathcal{N} = \{1, \ldots, n\}$ is the agent set. The environment has a global state space $\mathcal{S}$, and the joint action space is $\mathcal{A} = \prod_{i \in \mathcal{N}} \mathcal{A}_i$, with $\mathcal{A}_i$ the action set for agent $i$. At each timestep, every agent selects an action $a_i \in \mathcal{A}_i$, forming a joint action $\mathbf{a} = (a_1, a_2, \ldots, a_n) \in \mathcal{A}$. The transition dynamics are governed by $P(\mathbf{s}' \mid \mathbf{s}, \mathbf{a})$, and the global reward function is $R(\mathbf{s}, \mathbf{a})$. In partially observable settings, each agent receives a local observation $o_i \in \mathcal{O}_i$ via the observation function $\mathcal{Z}_i(\mathbf{s})$, with the joint observation denoted by $\mathbf{o} = (o_1, \ldots, o_n)$. In practice, the true global state $\mathbf{s}$ is not accessible. *For notational simplicity, we continue to use $\mathbf{s}$ in the formulation, though in implementation it corresponds to $\mathbf{o}$.* The objective is to learn a joint policy $\boldsymbol{\pi} = \{\pi_1, \ldots, \pi_n\}$ that maximizes the expected discounted return:

$$\max_{\boldsymbol{\pi}} \ \mathbb{E}_{\{\mathbf{s}_t, \mathbf{a}_t\} \sim \boldsymbol{\pi}} \left[ \sum_{t=0}^{\infty} \gamma^t R(\mathbf{s}_t, \mathbf{a}_t) \right].$$

Our work focuses on the setting of cooperative MARL under *sparse rewards*, where agents receive learning signals only at the trajectory level. Formally, for any trajectory (or episode) $\sigma$, the return is observed only upon termination: $R(\sigma) = \sum_{(\mathbf{s}_t, \mathbf{a}_t) \in \sigma} \gamma^t R(\mathbf{s}_t, \mathbf{a}_t)$, i.e., agents only receive episodic feedback. In this sparse-reward setting, intermediate timesteps provide no informative feedback, and agents must discover effective coordination strategies solely based on delayed, episodic outcomes. This poses two fundamental difficulties: *temporal credit assignment*, where the learning algorithm must infer which joint actions along the trajectory contributed to the final outcome, and *multi-agent credit assignment*, where responsibility for success must be attributed across multiple agents.

# 4 PREFERENCE-GUIDED IMPLICIT REWARD RECOVERY

As discussed earlier, the sparse-reward setting is particularly challenging in MARL, as the contribution of any single agent to the final episodic reward is entangled with the collective behavior of the entire team, significantly amplifying the difficulty of both temporal and multi-agent credit assignment. A conventional approach to mitigating the sparse-reward issue is to first *recover denser transition-level*

*rewards* from the trajectory-level signal and then standard MARL algorithms can then be applied to learn policies accordingly. Typically, this involves framing reward recovery as a supervised learning problem — for example, by learning local rewards that minimize the squared error between predicted transition-level rewards and the observed trajectory return. However, such methods often ignore environment dynamics and inter-agent interactions, limiting their effectiveness in cooperative settings.

Our work proposes a new framework that leverages *(inverse) preference learning* (IPL), where the goal is to infer latent reward signals that explain observed performance preferences between trajectories, rather than directly regressing against scalar episodic outcomes. By doing so, our approach naturally captures the structure of environment dynamics and the interactive nature of multi-agent cooperation, providing richer and more informative supervision for policy learning.

**Implicit Transition Reward Learning via IPL.** To apply IPL, we first convert the sparse reward signal into preference feedback. For any two trajectories $\sigma_1, \sigma_2$ sampled from the environment, let $R(\sigma_1)$ and $R(\sigma_2)$ denote their respective episodic rewards. We construct a preference pair $\sigma_1 \succ \sigma_2$ (trajectory $\sigma_1$ is preferred over $\sigma_2$) if $R(\sigma_1) \geq R(\sigma_2)$, and $\sigma_2 \succ \sigma_1$ otherwise. Let $\mathcal{P}$ denote the dataset generated from the environment, consisting of preference pairs $(\sigma_1, \sigma_2)$. The objective of preference-based reinforcement learning (PbRL) is to learn a joint reward function from $\mathcal{P}$.

A common approach in PbRL is to model preferences using the Bradley–Terry (BT) model (Bradley & Terry, 1952) which defines the probability of preferring $\sigma_1$ over $\sigma_2$ as follows:

$$P_R(\sigma_1 \succ \sigma_2) = \frac{\exp\big(\sum_{(\mathbf{s}_t,\mathbf{a}_t)\in\sigma_1} \gamma^t R(\mathbf{s}_t,\mathbf{a}_t)\big)}{\exp\big(\sum_{(\mathbf{s}_t,\mathbf{a}_t)\in\sigma_1} \gamma^t R(\mathbf{s}_t,\mathbf{a}_t)\big) + \exp\big(\sum_{(\mathbf{s}_t,\mathbf{a}_t)\in\sigma_2} \gamma^t R(\mathbf{s},\mathbf{a})\big)},$$

A direct approach to recovering $R(\mathbf{s}, \mathbf{a})$ is to maximize the likelihood of the observed preference data:

$$\max_r \mathcal{L}(r \mid \mathcal{P}) = \max_r \sum_{(\sigma_1,\sigma_2)\in\mathcal{P}} \ln P_R(\sigma_1 \succ \sigma_2).$$

Once $R(\mathbf{s}, \mathbf{a})$ are recovered, a MARL algorithm can then be applied to learn a cooperative policy.

A shortcoming of this explicit reward learning approach is that it does not fully account for the environment dynamics. The BT likelihood treats the learned reward purely as a function mapping state–action pairs to scalar values, without considering how actions influence the subsequent distribution of future states. Moreover, in multi-agent settings, this static treatment of rewards neglects the fact that agents' actions jointly affect the transition dynamics, making credit assignment across agents more difficult and potentially leading to misaligned or inconsistent learned policies.

To address this drawback, we leverage the IPL framework (Hejna & Sadigh, 2024) to transform the direct reward learning formulation into one that operates in the $Q$-function space. Specifically, by rearranging the soft Bellman equation, we obtain the so-called *inverse soft Bellman operator*: $(\mathcal{T}^* Q_{\text{tot}})(\mathbf{s}, \mathbf{a}) = Q_{\text{tot}}(\mathbf{s}, \mathbf{a}) - \gamma \mathbb{E}_{\mathbf{s}'\sim P(\cdot|\mathbf{s},\mathbf{a})} V_{\text{tot}}(\mathbf{s}')$, where $Q_{\text{tot}}$ denotes the soft global $Q$-function, and $V_{\text{tot}}$ is the corresponding soft global value function given by the log-sum-exp over joint actions: $V_{\text{tot}}(\mathbf{s}) = \beta \log[\sum_{\mathbf{a}} \exp(\frac{Q_{\text{tot}}(\mathbf{s},\mathbf{a})}{\beta})]$, with $\beta > 0$ serving as the temperature parameter for the soft Bellman operator. Note that in the Bradley-Terry model, we fix the temperature $\tau = 1$ for identifiability, allowing $\beta$ to control the entropy regularization of the recovered policy. An important observation here is that the inverse Bellman operator establishes a one-to-one mapping between $Q_{\text{tot}}$ and the transition reward function, i.e., $R(\mathbf{s}, \mathbf{a}) = (\mathcal{T}^* Q_{\text{tot}})(\mathbf{s}, \mathbf{a})$. This allows us to directly learn the global $Q$-function with the training objective:

$$\mathcal{L}(Q_{\text{tot}} \mid \mathcal{P}) = \sum_{(\sigma_1,\sigma_2)\in\mathcal{P}} \ln P_{(\mathcal{T}^* Q_{\text{tot}})}(\sigma_1 \succ \sigma_2) + \sum_{(\mathbf{s}_t,\mathbf{a}_t)} \phi(\gamma^t \mathcal{T}^* Q_{\text{tot}})(\mathbf{s}_t,\mathbf{a}_t)) \tag{1}$$

where $R(\cdot)$ is replaced with $(\mathcal{T}^* Q_{\text{tot}})(\cdot)$ and $\phi(.)$ is a concave regularizer used to stabilize the training.

**Value Decomposition.** To make learning practical and efficient in multi-agent settings, the CTDE paradigm with value factorization is typically employed. In our context, this can be achieved by factorizing $Q_{\text{tot}}$ and $V_{\text{tot}}$ as mixtures of local value functions. Specifically, let $\mathbf{q}(\mathbf{s}, \mathbf{a}) = \{q_i(s_i, a_i) \mid i \in \mathcal{N}\}$ and $\mathbf{v}(\mathbf{s}) = \{v_i(s_i) \mid i \in \mathcal{N}\}$ denote the sets of local $Q$-functions and $V$-functions, respectively. We then represent $Q_{\text{tot}}$ and $V_{\text{tot}}$ as linear mixtures of these local functions:

$$Q_{\text{tot}}(\mathbf{s}, \mathbf{a}) = \mathcal{M}_w[\mathbf{q}](\mathbf{s}, \mathbf{a}) = \sum_{i\in\mathcal{N}} w_i \, q_i(s_i, a_i) + w, \tag{2}$$

$$V_{\text{tot}}(\mathbf{s}) = \mathcal{M}_w[\mathbf{v}](\mathbf{s}) = \sum_{i\in\mathcal{N}} w_i \, v_i(s_i) + w, \tag{3}$$

where $\mathcal{M}_w[\cdot]$ denotes a linear mixing network parameterized by weights $\{w_i\}$. Here, $w_i$ denotes the contribution weight of each agent $i$ to the global value functions. Thus, for consistency, we employ the same set of parameters $\{w_i\}$ for both $Q_{\text{tot}}$ and $V_{\text{tot}}$. Moreover, we adopt a *linear mixing network* for both global value functions, motivated by the empirically strong performance of linear architectures reported in recent works Bui et al. (2025b;a). The linear formulation also facilitates an explicit *additive structure* of the global reward function, which in turn enables the construction of our dual-advantage PPO algorithm introduced in the next section. In contrast, nonlinear mixing architectures such as QMIX can provide more flexible representations but often suffer from overfitting (Bui et al., 2025b) and do not preserve the additive decomposition required for our dual-advantage policy optimization framework.

The training objective can thus be expressed in terms of the local functions $\mathbf{q}$ and $\mathbf{v}$ by substituting $Q_{\text{tot}}$ and $V_{\text{tot}}$ with their linear decompositions, yielding the preference-based loss $\mathcal{L}(\mathbf{q}, \mathbf{v} \mid \mathcal{P})$. To ensure that the global $V_{\text{tot}}$ satisfies the log-sum-exp consistency condition, we further update the local $V$-functions by minimizing the following *extreme-V* (Garg et al., 2023) objective defined under the mixing structures:

$$\mathcal{J}(\mathbf{v}|\mathbf{q}) = \mathbb{E}_{(\mathbf{s},\mathbf{a})}\left[\exp\left(\tfrac{\mathcal{M}_w[\mathbf{q}(\mathbf{s},\mathbf{a})]-\mathcal{M}_w[\mathbf{v}(\mathbf{s})]}{\beta}\right)\right] - \mathbb{E}_{(\mathbf{s},\mathbf{a})}\left[\tfrac{\mathcal{M}_w[\mathbf{q}(\mathbf{s},\mathbf{a})]-\mathcal{M}_w[\mathbf{v}(\mathbf{s})]}{\beta}\right] - 1. \quad (4)$$

The learning procedure can be carried out in two alternating steps. At each iteration, we first update the local $Q$-functions $\{q_i\}$ by maximizing the preference-based objective $\mathcal{L}(\mathbf{q}, \mathbf{v} \mid \mathcal{P})$ derived in the previous section. This step ensures that the learned $Q_{\text{tot}}$, expressed as a linear mixture of local components, is consistent with the observed trajectory preferences. Next, for each fixed set of local $Q$-functions $\mathbf{q}$, we update the local $V$-functions $\{v_i\}$ by minimizing the convex surrogate objective $\mathcal{J}(\mathbf{v} \mid \mathbf{q})$. This step enforces the consistency condition that the global value function $V_{\text{tot}}$ converges to the log-sum-exp of $Q_{\text{tot}}$, thereby ensuring that the implicit soft Bellman structure is preserved.

This implicit reward-learning framework, when combined with value decomposition, naturally incorporates both the environment dynamics and the inter-agent dependencies present in cooperative MARL. By working in the $Q$-space, the method provides stable gradients for policy optimization, leading to more reliable and effective policy learning under sparse-reward conditions.

**Theoretical Analysis.** Our following analysis shows that, under suitable conditions on the coverage of trajectory samples, and the specification of preference feedback, the recovered implicit reward will converge almost surely to a set of reward functions that includes the ground-truth reward up to a constant shift. To start, let $R^*(\mathbf{s}, \mathbf{a})$ be the ground-truth global transition reward. We define the following set which contains all reward functions whose trajectory-level return differs from that of the ground-truth reward only by an additive constant:

$$\mathcal{R} = \left\{ R(\mathbf{s}, \mathbf{a}) \mid \exists c \in \mathbb{R}, \sum_{(\mathbf{s}_t,\mathbf{a}_t)\in\sigma} \gamma^t R(\mathbf{s}_t, \mathbf{a}_t) = \sum_{(\mathbf{s}_t,\mathbf{a}_t)\in\sigma} \gamma^t R^*(\mathbf{s}_t, \mathbf{a}_t)+c, \ \forall \text{ trajectories } \sigma \right\},$$

An important property of this equivalence class is that any reward function in $\mathcal{R}$ yields the same optimal policy as the one defined by the ground-truth reward $R^*$ (Ng et al., 1999). Next, we show that under suitable conditions, if preference feedback is generated explicitly according to the BT model, then solving the preference-based objective $\max_{Q_{\text{tot}}} \mathcal{L}(Q_{\text{tot}} \mid \mathcal{P})$ yields an implicit reward $R(\mathbf{s}, \mathbf{a}) = Q_{\text{tot}}(\mathbf{s}, \mathbf{a}) - \gamma V_{\text{tot}}(\mathbf{s})$, which converges asymptotically to some $\widetilde{R} \in \mathcal{R}$.

**Theorem 1** (Asymptotic Convergence). *Assume that preference feedback is generated according to the BT model with inverse temperature $\tau = 1$. That is, for two trajectories $\sigma_1, \sigma_2$, define noisy utilities $U(\sigma_1) = R^*(\sigma_1) + \epsilon_1$, $U(\sigma_2) = R^*(\sigma_2) + \epsilon_2$, where $\epsilon_1, \epsilon_2$ are i.i.d. Gumbel-distributed random variables. Suppose that $\mathcal{P}$ contains every possible preference pair $(\sigma_1, \sigma_2)$ (i.e., $U(\sigma_1) \geq U(\sigma_2)$), each observed at least $N$ times. Then, as $N \to \infty$, the recovered implicit reward $R(\boldsymbol{s}, \boldsymbol{a})$ will asymptotically matches the ground-truth reward $R^*$ up to an additive constant at the trajectory level.*

We note that in the above theorem, we introduce Gumbel-distributed noise to the aggregated trajectory rewards. This makes the preference between two trajectories $\sigma_1$ and $\sigma_2$ stochastic, with probability following the Bradley–Terry model. Such probabilistic labeling ensures that the induced preference distribution aligns with the BT likelihood, which is crucial for establishing the *asymptotic convergence*. Importantly, this stochastic assumption serves only a theoretical role, as our practical implementation does not inject explicit noise but naturally approximates this behavior through inherent environment and sampling randomness.

Theorem 1 together highlight the robustness of the preference-based learning framework. These results imply that preference-based reward learning is guaranteed to converge to a behaviorally indistinguishable surrogate of the ground-truth reward. This provides a solid theoretical foundation for our framework, ensuring that—even though the exact reward may not be uniquely identifiable—the induced optimal policies remain aligned with those of the true underlying objective. While Theorem 1 guarantees asymptotic convergence (i.e., convergence holds only with sufficiently large sample sizes, which is rarely achievable in practice), it nonetheless provides theoretical insight into the convergence of our preference-based algorithm toward the true reward function. In practice, we adopt an iterative online learning process, maintaining a FIFO preference buffer $\mathcal{P}$ so that the reward model continuously adapts to the evolving state distribution induced by the current policy, thereby mitigating the finite-sample gap.

**Enhancing Preference Learning with LLMs.** While PbRL provides a principled way to infer implicit reward functions from sparse trajectory-level feedback, relying solely on trajectory-level cumulative rewards to determine preferences can be restrictive. In many domains, the scalar outcome signal may not fully capture nuanced aspects of trajectory quality. For instance, in the SMAC benchmark, two trajectories may achieve similar accumulated rewards even though one trajectory clearly exhibits more desirable cooperative behavior (e.g., coordinated unit positioning, efficient focus fire, or minimal resource wastage), while the other relies on risky or inefficient strategies.

This motivates the integration of LLMs into the preference-learning framework. LLMs are well-suited to process entire trajectories, which are structured sequences of state–action pairs augmented with trajectory-level statistics. The key advantage of using LLMs lies in their ability to consider both quantitative outcomes (e.g., accumulated rewards, win/loss indicators) and qualitative, interpretable features extracted from the trajectory. For example, in SMAC, an LLM can be prompted to evaluate whether agents maintained formation, avoided unnecessary deaths, or executed coordinated maneuvers. These features can be expressed as natural-language descriptions or structured indicators, which the LLM can weigh alongside scalar rewards to generate more informed preference judgments.

Technically, trajectories can be encoded into descriptive summaries using domain-specific features (such as unit health, spatial coverage, or coordination metrics) combined with raw outcome statistics (such as cumulative reward or terminal success signals). These summaries are provided as prompts to the LLM, which outputs preference scores or pairwise comparisons between trajectories. In this way, the LLM serves as an auxiliary judge, complementing reward-derived preferences with semantically enriched, context-aware assessments. This augmentation would yield preference data that are less noisy and more aligned with high-level performance criteria. Furthermore, the integration of interpretable features provides transparency: instead of merely inferring that one trajectory is "better" than another, the LLM can articulate *why* a trajectory is preferred (e.g., agents maintained formation while minimizing casualties). Such interpretability can effectively facilitate human-in-the-loop training, where expert feedback can be incorporated more seamlessly.

## 5 MULTI-AGENT PPO WITH DUAL ADVANTAGE STREAMS

### 5.1 STANDARD MAPPO WITH PREFERENCE-DERIVED REWARDS

Given the recovered $R(\mathbf{s}, \mathbf{a})$, we now discuss the MAPPO algorithm (Yu et al., 2022) that can be employed to learn decentralized policies accordingly. Let $\pi_{\theta_i}(a_{i,t} \mid s_{i,t})$ denote the local policy for agent $i$, where $s_{i,t}$ is agent $i$'s local state at time $t$ and $\mathbf{s}_t = (s_{1,t}, \ldots, s_{n,t})$. A centralized critic $V_\phi^{\text{tot}}(\mathbf{s}_t)$ is trained using the global information, while each agent $i$ maintains a decentralized actor $\pi_{\theta_i}(a_{i,t} \mid s_{i,t})$ that only consumes its local observation during both training and execution.

MAPPO employs generalized advantage estimation (GAE) with the preference-derived reward:

$$\Delta_t^{\text{tot}} = R(\mathbf{s}_t, \mathbf{a}_t) + \gamma V_\phi^{\text{tot}}(\mathbf{s}_{t+1}) - V_\phi^{\text{tot}}(\mathbf{s}_t), \quad \hat{A}_t^{\text{tot}} = \sum_{l=0}^{\infty} (\gamma \lambda_{\text{GAE}})^l \Delta_{t+l}^{\text{tot}},$$

where $\gamma \in [0, 1)$ is the discount factor and $\lambda_{\text{GAE}} \in [0, 1]$ controls the bias–variance trade-off.
**Decentralized actors:** For each agent $i$, let's define the importance ratio $\rho_{i,t}(\boldsymbol{\theta}) = \frac{\pi_{\theta_i}(a_{i,t}|s_{i,t})}{\pi_{\theta_i^{\text{old}}}(a_{i,t}|s_{i,t})}$.

MAPPO trains decentralized actors by maximizing the clipped surrogate with an entropy bonus:

$$\mathcal{L}_{\text{actor}}(\boldsymbol{\theta}) = \mathbb{E}_t\Big[ \sum_{i=1}^{n} \min\Big( \rho_{i,t}(\boldsymbol{\theta}) \, \hat{A}_t^{\text{tot}}, \, \text{clip}\big( \rho_{i,t}(\boldsymbol{\theta}), \, 1-\epsilon, \, 1+\epsilon \big) \hat{A}_t^{\text{tot}} \Big) + \eta \, \mathcal{H}\big( \pi_{\theta_i}(\cdot \mid o_{i,t}) \big) \Big], \quad (5)$$

where $\epsilon > 0$ is the PPO clipping parameter, $\eta \geq 0$ weights the entropy regularizer $\mathcal{H}$.

**Centralized Critic:** Let the empirical return target be $\hat{R}_t = \hat{A}_t^{\mathrm{tot}} + V_{\phi_{\mathrm{old}}}^{\mathrm{tot}}(\mathbf{s}_t)$. MAPPO trains a centralized critic with the following clipped value loss:

$$\mathcal{L}_{\mathrm{critic}}(\phi) \;=\; \mathbb{E}_t\big[\max\big(\big(V_\phi^{\mathrm{tot}}(\mathbf{s}_t) - \hat{R}_t\big)^2, \; \big(\hat{V}_t^{\mathrm{clip}} - \hat{R}_t\big)^2\big)\big], \tag{6}$$

where $\hat{V}_t^{\mathrm{clip}} = \mathrm{clip}(V_\phi^{\mathrm{tot}}(\mathbf{s}_t), \; V_{\phi_{\mathrm{old}}}^{\mathrm{tot}}(\mathbf{s}_t) - \epsilon_v, \; V_{\phi_{\mathrm{old}}}^{\mathrm{tot}}(\mathbf{s}_t) + \epsilon_v)$ and $\epsilon_v > 0$ is the clipping parameter.

In summary, the training alternates between (i) collecting trajectories under $\pi_{\boldsymbol{\theta}}$, (ii) computing the global advantage $\hat{A}_t^{\mathrm{tot}}$ via GAE using the implicit reward $R(\mathbf{s}, \mathbf{a})$, and (iii) performing stochastic gradient updates that maximize equation 5 and minimize equation 6. A key *shortcoming* of this standard MAPPO formulation is that all decentralized actors are trained using the *same* global advantage estimate $\hat{A}_t^{\mathrm{tot}}$. This design ignores the heterogeneity of local agents and their individual contributions to the joint return. Consequently, the training signal for each local actor fails to capture *agent-specific credit assignment*, which can lead to inefficient updates and suboptimal convergence in cooperative settings where different agents have asymmetric responsibilities.

## 5.2 MAPPO WITH DUAL ADVANTAGE STREAMS

To address the above issue of standard MAPPO, we introduce a novel approach that enables learning decentralized policies and a centralized critic with both local and global advantage functions, leveraging the information obtained from the implicit reward learning phase.

**Global and Local Advantage Estimates.** Recall that the reward recovery framework provides both local $Q$- and $V$-functions, and the weights $\{w_i^*, w^*\}$ of the mixing network. These weights capture the inter-dependencies among agents, describing how local components contribute to the global value functions. Hence, we can recover both the global implicit reward and agent-specific local rewards. Formally, the global implicit reward can be computed via the inverse soft Bellman operator: $R(\mathbf{s}, \mathbf{a}) = \widetilde{Q}_{\mathrm{tot}}(\mathbf{s}, \mathbf{a}) - \gamma \, \mathbb{E}_{\mathbf{s}' \sim P(\cdot|\mathbf{s}, \mathbf{a})}\big[\widetilde{V}_{\mathrm{tot}}(\mathbf{s}')\big]$, where $\widetilde{Q}_{\mathrm{tot}}$ and $\widetilde{V}_{\mathrm{tot}}$ denote the global $Q$- and $V$-functions obtained from the preference-learning step. Note that we do not directly reuse the global value functions $\widetilde{Q}_{\mathrm{tot}}$ and $\widetilde{V}_{\mathrm{tot}}$ obtained from the preference-learning stage for policy extraction. These value estimates are learned through a weak interaction with the environment and may therefore lack the accuracy required to effectively guide policy optimization. As shown in our experiments, the baseline Online-IPL, which relies on these global values for policy extraction, exhibits significantly lower sample efficiency. Instead, we use the reward function implicitly derived from $\widetilde{Q}_{\mathrm{tot}}$ and $\widetilde{V}_{\mathrm{tot}}$, which captures the relative preference between state–action pairs. This reward signal is then integrated into the MAPPO framework, allowing the value functions and policies to be refined through richer and more stable interactions with the environment dynamics.

For each agent $i$ we define the local implicit reward as function of the local Q and V functions: $r_i(s_i, a_i) = \widetilde{q}_i(s_i, a_i) - \gamma \, \mathbb{E}_{s_i' \sim P(\cdot|\mathbf{s}_i, \mathbf{a}_i)}\big[\widetilde{v}_i(s_i')\big]$. Assuming that both $\widetilde{Q}_{\mathrm{tot}}$ and $\widetilde{V}_{\mathrm{tot}}$ share the same linear mixing network with weights $\{w_i^*\}$, we can express the global reward as a linear combination of the local rewards: $R(\mathbf{s}, \mathbf{a}) = \sum_{i=1}^{n} w_i^* r_i(s_i, a_i) + (1 - \gamma)w^*$. To incorporate dual advantage streams into the CTDE paradigm, we define the following *local advantage function* for each agent $i \in \mathcal{N}$:

$$\delta_{i,t}^{\mathrm{local}} = r_i(s_{i,t}, a_{i,t}) + \frac{\gamma}{w_i^*}\big(V_\phi^{\mathrm{tot}}(\mathbf{s}_{t+1}) - V_\phi^{\mathrm{tot}}(\mathbf{s}_t)\big); \quad \hat{A}_{i,t}^{\mathrm{local}} = \sum_{l=0}^{\infty}(\gamma\lambda_{\mathrm{GAE}})^l \, \delta_{i,t+l}^{\mathrm{local}},$$

**Proposition 1.** *The global advantage at time $t$ satisfies* $\hat{A}_t^{\mathrm{tot}} \;=\; \sum_{i \in \mathcal{N}} w_i^* \, \hat{A}_{i,t}^{\mathrm{local}} \;+\; (1 - \gamma)\, w^*$, *where* $\{w_i^*\}$ *and* $w^*$ *are the weights and bias term of the mixing networks.*

**MAPPO with Dual Advantage Streams.** With the local advantage functions defined above, we modify the MAPPO decentralized actor objective to incorporate agent-specific information as follows:

$$\mathcal{L}_{\mathrm{actor}}^{\mathrm{dual}}(\boldsymbol{\theta}) = \mathbb{E}_t\Big[\sum_{i=1}^{n} \min\big(\rho_{i,t}(\boldsymbol{\theta})\,\hat{A}_{i,t}^{\mathrm{local}}, \mathrm{clip}\big(\rho_{i,t}(\boldsymbol{\theta}), 1-\epsilon, 1+\epsilon\big)\,\hat{A}_{i,t}^{\mathrm{local}}\big) + \eta\mathcal{H}\big(\pi_{\theta_i}(\cdot|s_{i,t})\big)\Big]. \tag{7}$$

The centralized critic follows the same design as in the original MAPPO, using the global advantage estimate $\hat{A}_t^{\mathrm{tot}}$. The key advantage of this decentralized actor formulation is that each agent updates its

policy using *local advantage signals* that directly reflect its individual contribution to the team's outcome. Our dual-stream formulation disentangles global and local contributions: the centralized critic benefits from global coordination, while the decentralized actors leverage local credit assignments to achieve more accurate, stable, and sample-efficient policy updates in sparse-reward settings.

**Global–Local Consistency.** The use of local advantages above can be interpreted as a form of value factorization, where the global advantage is decomposed into local components to facilitate decentralized learning. A key question in this approach is whether the local policy updates induced by these agent-specific advantages remain *consistent* with the optimization of the underlying global joint policy. This concern has been widely studied in prior MARL work on value decomposition, but here we revisit it in the context of preference-based implicit rewards and dual advantage streams.

Let us recall the joint policy optimization formulation underlying standard MAPPO: $\nabla_{\boldsymbol{\theta}} J(\boldsymbol{\pi_\theta}) = \mathbb{E}_t\big[\nabla_{\boldsymbol{\theta}} \log \boldsymbol{\pi_\theta}(\mathbf{a}_t \mid \mathbf{s}_t) \, \hat{A}_t^{\text{tot}}(\mathbf{s}_t, \mathbf{a}_t)\big]$, whereas the local policy optimization for each local agent $i$, using its local advantage: $\nabla_{\theta_i} J(\pi_{\theta_i}) = \mathbb{E}_t\big[\nabla_{\theta_i} \log \pi_{\theta_i}(a_{i,t} \mid s_{i,t}) \, \hat{A}_{i,t}^{\text{local}}(s_{i,t}, a_{i,t})\big]$,

**Proposition 2.** *Assume that the joint policy factorizes as $\boldsymbol{\pi_\theta}(\mathbf{a}_t \mid \mathbf{s}_t) = \prod_{i=1}^n \pi_{\theta_i}(a_{i,t} \mid s_{i,t})$, then: $\nabla_{\boldsymbol{\theta}} J(\boldsymbol{\pi_\theta}) = \sum_{i=1}^n w_i^* \, \nabla_{\theta_i} J(\pi_{\theta_i})$, where $w_i^*$ are the mixing weights of the preference learning.*

Prop. 2 establishes that optimizing local policies w.r.t their agent-specific advantages is *consistent* with optimizing the global joint policy. That is, the joint policy gradient can be decomposed into a weighted sum of the local policy gradients, ensuring that decentralized updates remain aligned with the global objective. This result highlights a key property of our framework: by grounding local updates in the dual advantage stream, we provide each agent with an individualized yet globally consistent learning signal, avoiding the credit misattribution problem in standard MAPPO.

**Practical Implementation:** Our proposed framework, IMAP (**I**nverse Preference-Guided **M**ulti-**A**gent **P**olicy Optimization), integrates online IPL with a PPO-style algorithm to address sparse-reward MARL. The practical implementation is detailed in appendix. At each iteration, the algorithm first collects a batch of trajectories using the current policies. These trajectories are then used to generate preference pairs. In the rule-based version, a preference $(\sigma_i \succ \sigma_j)$ is created if the final sparse reward of trajectory $\sigma_i$ is greater than that of $\sigma_j$, i.e., $R(\sigma_i) > R(\sigma_j)$. For tasks with interpretable features (e.g., SMAC), detailed trajectory summaries are provided to a LLM, which in turn produces more nuanced preference judgments based on high-level strategic objectives. Once preferences are collected, the implicit reward model is updated. This involves training the local Q-networks and the mixing network to maximize the preference log-likelihood, and updating the local V-networks to maintain consistency between the global and local value functions. Finally, the actor and critic networks are updated using the PPO algorithm. The implicit rewards derived from the learned value functions are used to compute dual advantage estimates: a global advantage for the centralized critic and agent-specific local advantages for the decentralized actors. The entire process creates a synergistic loop where policy improvement and reward refinement occur in tandem.

# 6 EXPERIMENTS

We evaluate IMAP on two challenging multi-agent benchmarks: the StarCraft II Multi-Agent Challenge (SMACv2) (Ellis et al., 2023) for discrete control and Multi-Agent MuJoCo (MaMujoco) (de Witt et al., 2020) for continuous control. We compare against strong baselines, including **SparseMAPPO**, which applies MAPPO directly to trajectory-level rewards; **SparseHAPPO**, which applies HAPPO (Kuba et al., 2021) (a heterogeneous-agent variant of MAPPO) directly to trajectory-level rewards; **SL-MAPPO**, which first recovers transition-level rewards and then trains policies with MAPPO; and **Online-IPL**, our online inverse preference learning approach where global value functions $Q_{\text{tot}}$ and $V_{\text{tot}}$ are iteratively refined and combined with behavior cloning to extract local policies. Details of these baselines are provided in appendix. As part of our ablation studies, we evaluate two configurations for preference elicitation: (i) **IMAP-Rule**, which derives rule-based preferences from trajectory-level reward feedback, and (ii) **IMAP-LLM**, which leverages preferences generated by the Qwen-4B model (Qwen, 2025) (applicable to SMACv2 only)[1]. Note that some attention-based methods have been proposed to address sparse-reward MARL (She et al., 2022; Xiao

---

[1]Qwen-4B is lightweight and supports local querying, making it more suitable for online preference generation than larger models such as ChatGPT or Gemini.

Table 1: Performance comparison on SMACv2 scenarios. Results show mean win rate (%) $\pm$ standard deviation over 5 seeds.

| Algorithm | Protoss | | | Terran | | | Zerg | | |
|---|---|---|---|---|---|---|---|---|---|
| | 5_vs_5 | 10_vs_10 | 10_vs_11 | 5_vs_5 | 10_vs_10 | 10_vs_11 | 5_vs_5 | 10_vs_10 | 10_vs_11 |
| SparseMAPPO | $0.0 \pm 0.0$ | $0.0 \pm 0.0$ | $0.0 \pm 0.0$ | $32.0 \pm 3.3$ | $23.1 \pm 4.2$ | $0.0 \pm 0.0$ | $18.5 \pm 4.0$ | $10.4 \pm 2.5$ | $4.6 \pm 1.6$ |
| SparseHAPPO | $11.6 \pm 3.0$ | $0.0 \pm 0.0$ | $0.0 \pm 0.0$ | $35.5 \pm 3.6$ | $23.5 \pm 2.9$ | $0.0 \pm 0.0$ | $23.0 \pm 3.9$ | $13.3 \pm 2.0$ | $4.8 \pm 1.6$ |
| Online-IPL | $15.0 \pm 3.8$ | $9.4 \pm 2.5$ | $7.5 \pm 2.3$ | $32.5 \pm 4.4$ | $23.4 \pm 2.7$ | $10.7 \pm 1.8$ | $28.2 \pm 3.5$ | $14.4 \pm 2.3$ | $4.8 \pm 1.9$ |
| SL-MAPPO | $27.0 \pm 6.0$ | $21.4 \pm 3.6$ | $15.7 \pm 3.5$ | $38.1 \pm 5.6$ | $28.2 \pm 4.2$ | $14.7 \pm 2.5$ | $25.8 \pm 3.7$ | $21.0 \pm 1.4$ | $8.7 \pm 2.7$ |
| IMAP-Rule | $44.0 \pm 3.2$ | $33.6 \pm 2.8$ | $23.1 \pm 4.0$ | $46.8 \pm 2.9$ | $38.2 \pm 2.8$ | $19.7 \pm 2.1$ | $42.1 \pm 3.4$ | $27.8 \pm 3.7$ | $17.2 \pm 1.8$ |
| IMAP-LLM | $\mathbf{48.3 \pm 2.8}$ | $\mathbf{37.8 \pm 2.3}$ | $\mathbf{25.2 \pm 4.7}$ | $\mathbf{55.5 \pm 3.1}$ | $\mathbf{38.1 \pm 3.1}$ | $\mathbf{26.3 \pm 4.6}$ | $\mathbf{43.8 \pm 5.1}$ | $\mathbf{29.8 \pm 2.8}$ | $\mathbf{18.4 \pm 2.2}$ |

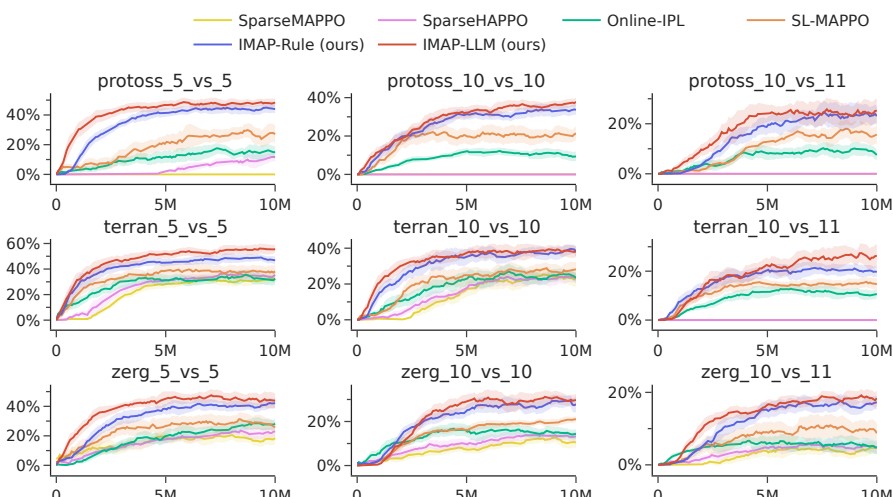

Figure 1: Learning curves (winrates) on nine SMACv2 scenarios.

et al., 2022; Chen et al., 2024; Kapoor et al., 2025), but they are generally less scalable and struggle to handle challenging benchmarks such as SMACv2 or MAMuJoCo. For completeness, we include a comparison with these baselines in Appendix C.4, based on lightweight tasks from the SMACLite (Michalski et al., 2023).

**SMACv2.** In the SMACv2 scenarios, performance is measured by the mean win rate. As shown in Table 1, our IMAP framework significantly outperforms all baselines across all Protoss, Terran, and Zerg maps. SparseMAPPO fails to learn in most scenarios, highlighting the difficulty of the sparse-reward problem. SparseHAPPO exhibits modest gains over SparseMAPPO in certain scenarios (e.g., Protoss 5_vs_5 and Zerg tasks), suggesting that accounting for agent heterogeneity offers some advantage. However, it remains fundamentally limited by the sparsity of the reward signal. While Online-IPL and SL-MAPPO show further improvement by learning an explicit reward, they are consistently surpassed by our implicit reward learning approach. IMAP-Rule already achieves strong performance, and IMAP-LLM further boosts the win rates, demonstrating the value of semantically rich preference feedback from LLMs for complex coordination tasks. The learning curves in Figure 6 and the summary box plots in Figure 5 clearly illustrate the superior performance and sample efficiency of our methods.

**MaMujoco.** In the continuous control tasks of MaMujoco, our IMAP-Rule method again demonstrates superior performance, as shown in Table 9. It achieves substantially higher total rewards than all baselines across all four environments. SparseHAPPO achieves slight improvements over SparseMAPPO in tasks such as Hopper-v2 and Swimmer-v2, but the gains are marginal, and both methods are significantly outperformed by reward learning approaches. For example, in HalfCheetah-v2, IMAP obtains a score more than double that of SparseMAPPO and SparseHAPPO. This shows that our implicit reward learning mechanism is general and adapts effectively to continuous action spaces without requiring LLM guidance. We note that the continuous-control environments MAMu-JoCo lack such interpretable features, making LLM-based evaluation unreliable. Therefore, we report

only the IMAP-Rule variant for MAMuJoCo, which provides a fairer and more stable comparison in this domain.

Table 2: Performance on MAMuJoCo tasks. Results show mean total episode reward ± std.

| Algorithm | Hopper-v2 | Reacher-v2 | HalfCheetah-v2 | Swimmer-v2 |
|---|---|---|---|---|
| SparseMAPPO | $188.6 \pm 18.3$ | $-488.9 \pm 76.1$ | $1824.3 \pm 181.5$ | $16.3 \pm 2.1$ |
| SparseHAPPO | $215.4 \pm 30.0$ | $-497.6 \pm 77.7$ | $1926.7 \pm 224.8$ | $20.5 \pm 1.8$ |
| Online-IPL | $243.5 \pm 9.7$ | $-359.2 \pm 17.3$ | $1777.6 \pm 341.4$ | $23.1 \pm 0.8$ |
| SL-MAPPO | $296.5 \pm 19.1$ | $-358.2 \pm 15.5$ | $2479.4 \pm 337.5$ | $29.8 \pm 1.2$ |
| IMAP-Rule | $\mathbf{361.2 \pm 16.2}$ | $\mathbf{-182.1 \pm 13.3}$ | $\mathbf{3853.6 \pm 359.0}$ | $\mathbf{34.0 \pm 1.0}$ |

**Computation Cost.** Querying Qwen-4B takes approximately 0.03s per preference pair on a single A100 GPU. While higher than scalar rewards, this cost is incurred only during training (reward labeling). Execution remains efficient as actors use lightweight GRUs.

**Ablation Studies.** We perform two ablation studies to assess the contributions of our framework. **First**, we compared the full model with local advantages (**IMAP-LA**) against a variant using only a shared global advantage (**IMAP-GA**). As reported in Tables 10 and 11, the dual-stream design consistently outperforms the single-stream variant, demonstrating that agent-specific credit assignment through local advantages is crucial for effective and efficient multi-agent learning. **Second**, we evaluate the role of different lightweight LLMs (Gemma3-270M, Gemma3-4B, and Qwen-4B) for preference generation on SMACv2. Our results (see Appendix for details) indicate that more capable LLMs yield stronger guidance, resulting in higher win rates.

Table 3: Comparison of two variants: **IMAP-LA** against **IMAP-GA**, on SMACv2 tasks.

| | Protoss | | Terran | | Zerg | |
|---|---|---|---|---|---|---|
| Scenario | IMAP-GA | IMAP-LA | IMAP-GA | IMAP-LA | IMAP-GA | IMAP-LA |
| 5_vs_5 | $43.52 \pm 2.21$ | $\mathbf{44.00 \pm 3.16}$ | $45.26 \pm 2.15$ | $\mathbf{46.84 \pm 2.93}$ | $38.48 \pm 5.48$ | $\mathbf{42.14 \pm 3.44}$ |
| 10_vs_10 | $31.71 \pm 2.19$ | $\mathbf{33.64 \pm 2.78}$ | $36.23 \pm 3.26$ | $\mathbf{38.18 \pm 2.82}$ | $26.15 \pm 1.51$ | $\mathbf{27.78 \pm 3.65}$ |
| 10_vs_11 | $22.01 \pm 3.75$ | $\mathbf{23.08 \pm 3.99}$ | $17.46 \pm 4.25$ | $\mathbf{19.71 \pm 2.08}$ | $14.42 \pm 2.80$ | $\mathbf{17.24 \pm 1.77}$ |

## 7 CONCLUSION

We proposed IMAP, a preference-guided framework for cooperative MARL under sparse rewards that unifies implicit reward learning with policy optimization via dual advantage streams, and provided rigorous theoretical analysis on the asymptotic convergence of preference learning as well as the global–local consistency of the dual-advantage policy optimization approach. Experiments on state-of-the-art benchmarks, SMACv2 and MAMuJoCo, demonstrate clear improvements in both performance and sample efficiency over strong baselines. Limitations of the current work include its restriction to cooperative settings, the reliance on interpretable environments for LLM-based preferences, and the high cost of online LLM queries, which limits the use of larger models such as ChatGPT. These challenges point to promising directions for future research.

## REPRODUCIBILITY STATEMENT

We have made substantial efforts to ensure the reproducibility of our work. The proposed IMAP framework is described in detail in Sections 4–5, with the full training pipeline outlined in Algorithm 1. All theoretical results are accompanied by precise assumptions and complete proofs, which can be found in Appendix A. For experimental reproducibility, we provide comprehensive descriptions of the environments (SMACv2 and MaMuJoCo), hyperparameter settings, and implementation details in Appendix B. Additional ablation studies and sensitivity analyses are also reported to verify robustness. Moreover, we include the source code in the submission and configuration files in the supplementary materials, allowing readers to directly replicate our experiments. Finally, the prompts and templates used for LLM-based preference labeling are documented in Appendix C, ensuring clarity in reproducing the preference generation procedure.

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

# A    MISSING PROOFS

We provide proofs that are omitted in the main paper.

## A.1    PROOF OF THEOREM 1

**Theorem 1.**    *Assume that preference feedback is generated according to the BT model with inverse temperature $\tau = 1$. That is, for two trajectories $\sigma_1, \sigma_2$, define noisy utilities $U(\sigma_1) = R^*(\sigma_1) + \epsilon_1$, $U(\sigma_2) = R^*(\sigma_2) + \epsilon_2$, where $\epsilon_1, \epsilon_2$ are i.i.d. Gumbel-distributed random variables. Suppose that $\mathcal{P}$ contains every possible preference pair $(\sigma_1, \sigma_2)$ (i.e., $U(\sigma_1) \geq U(\sigma_2)$), each observed at least $N$ times. Then, as $N \to \infty$, the recovered implicit reward $R(\boldsymbol{s}, \boldsymbol{a})$ will asymptotically matches the ground-truth reward $R^*$ up to an additive constant at the trajectory level.*

*Proof.*  We proceed in four steps: (i) population optimality of the BT likelihood, (ii) representation via the inverse soft Bellman operator, (iii) characterization of the equivalence class, and (iv) convergence of the empirical maximizer.

**Population preference-based likelihood:**    For any trajectory $\sigma$, define its cumulative reward under a transition reward function $R$ as

$$S_R(\sigma) = \sum_{(\mathbf{s}_t, \mathbf{a}_t) \in \sigma} \gamma^t R(\mathbf{s}_t, \mathbf{a}_t).$$

Under the BT model with fixed inverse temperature $\tau = 1$, the probability of observing $\sigma_1 \succ \sigma_2$ is

$$\Pr(\sigma_1 \succ \sigma_2) = \frac{\exp(S_{R^*}(\sigma_1))}{\exp(S_{R^*}(\sigma_1)) + \exp(S_{R^*}(\sigma_2))},$$

where $R^*$ is the ground-truth reward. Let $\Pi$ denote the distribution over trajectory pairs. The *population* log-likelihood of a candidate score function $S$ is

$$\mathcal{L}_\infty(S) = \mathbb{E}_{(\sigma_1, \sigma_2) \sim \Pi}\Big[\mathbf{1}\{\sigma_1 \succ \sigma_2\} \log \Delta(S(\sigma_1) - S(\sigma_2)) + \mathbf{1}\{\sigma_2 \succ \sigma_1\} \log \Delta(S(\sigma_2) - S(\sigma_1))\Big],$$

where $\Delta(u) = 1/(1 + e^{-u})$ is the logistic function. By strict concavity of the logistic log-likelihood in the score differences $\Delta S = S(\sigma_1) - S(\sigma_2)$, the maximizers of $\mathcal{L}_\infty$ are exactly those $S$ that match the true score differences of $S_{R^*}$. Hence

$$S(\sigma) = S_{R^*}(\sigma) + c, \qquad \forall \sigma,$$

for some additive constant $c \in \mathbb{R}$.

**Inverse soft Bellman operator:**    Rather than learning $R$ directly, recall that our method learns a global soft $Q$-function $Q_{\text{tot}}$ and its associated soft value function

$$V_{\text{tot}}(\mathbf{s}) = \beta \log \sum_{\mathbf{a}} \exp\Big(\tfrac{Q_{\text{tot}}(\mathbf{s}, \mathbf{a})}{\beta}\Big).$$

The implicit transition reward induced by $Q_{\text{tot}}$ is obtained via the inverse soft Bellman operator:

$$R_Q(\mathbf{s}, \mathbf{a}) = Q_{\text{tot}}(\mathbf{s}, \mathbf{a}) - \gamma \, \mathbb{E}_{\mathbf{s}' \sim P(\cdot|\mathbf{s}, \mathbf{a})}[V_{\text{tot}}(\mathbf{s}')].$$

For any trajectory $\sigma$, the trajectory score under $R_Q$ is

$$S_{R_Q}(\sigma) = \sum_{(\mathbf{s}_t, \mathbf{a}_t) \in \sigma} \gamma^t R_Q(\mathbf{s}, \mathbf{a}).$$

Thus optimizing over $Q_{\text{tot}}$ is equivalent to optimizing over a restricted class of trajectory score functions induced by such $R_Q$.

**Equivalence class $\mathcal{R}$:**    From above, we see that any population maximizer $S$ satisfies $S(\sigma) = S_{R^*}(\sigma) + c$. Therefore, the induced implicit reward $R_Q$ belongs to the equivalence class

$$\mathcal{R} = \Big\{R : \exists c \in \mathbb{R}, \ S_R(\sigma) = S_{R^*}(\sigma) + c, \ \forall \sigma\Big\}.$$

That is, $R_Q$ and $R^*$ differ only by a constant shift at the trajectory level, which does not affect policy optimization since advantages and relative preferences remain unchanged.

**Asymptotical convergence:** Let $\mathcal{L}_N(Q_{\text{tot}} \mid \mathcal{P})$ denote the empirical log-likelihood constructed from the dataset $\mathcal{P}$ of $N$ i.i.d. preference comparisons. Since the preference feedback is generated directly from the BT model (the true distribution) via Gumbel random noise, standard M-estimation arguments (van der Vaart, 1998) apply. Under mild regularity conditions (including ergodicity of the sampling process, sufficient coverage of trajectory pairs, and expressivity of the function class), we obtain uniform convergence $\mathcal{L}_N \to \mathcal{L}_\infty$. Moreover, because $\mathcal{L}_\infty$ is strictly concave in the differences $\Delta S$, the set of population maximizers is unique up to an additive constant. By the argmax consistency theorem, any sequence of empirical maximizers $\widehat{Q}_{\text{tot},N}$ converges in probability to a population maximizer. Consequently, the induced implicit reward

$$\widehat{R}_N(\mathbf{s}, \mathbf{a}) = \widehat{Q}_{\text{tot},N}(\mathbf{s}, \mathbf{a}) - \gamma \, \mathbb{E}_{\mathbf{s}'|\mathbf{s},\mathbf{a}}[V_{\widehat{Q}_{\text{tot},N}}(\mathbf{s}')]$$

converges in probability to the set $\mathcal{R}$.

Thus, in summary, the preference-based estimator recovers an implicit reward that asymptotically matches the ground-truth $R^*$ up to an additive constant at the trajectory level. We complete the proof. $\qquad\square$

### A.2 PROOF OF PROPOSITION 1

**Proposition 1.** *The global advantage at time $t$ satisfies $\hat{A}_t^{tot} = \sum_{i \in \mathcal{N}} w_i^* \hat{A}_{i,t}^{local} + (1-\gamma)\,w^*$, where $\{w_i^*\}$ and $w^*$ are the mixing weights and bias term returned by the preference-based reward decomposition.*

*Proof.* By definition of the implicit global reward recovered from preference learning:

$$R(\mathbf{s}, \mathbf{a}) = \widetilde{Q}_{\text{tot}}(\mathbf{s}, \mathbf{a}) - \gamma \, \mathbb{E}_{\mathbf{s}' \sim P(\cdot|\mathbf{s},\mathbf{a})}\big[\widetilde{V}_{\text{tot}}(\mathbf{s}')\big], \tag{8}$$

we can write:

$$R(\mathbf{s}_t, \mathbf{a}_t) = \sum_{i \in \mathcal{N}} w_i^* \, r_i(s_{i,t}, a_{i,t}) + (1-\gamma)\,w^*.$$

Substituting this decomposition into the temporal-difference error for the global critic yields

$$\delta_t^{\text{tot}} = R(\mathbf{s}_t, \mathbf{a}_t) + \gamma V_\phi^{\text{tot}}(\mathbf{s}_{t+1}) - V_\phi^{\text{tot}}(\mathbf{s}_t).$$

Replacing $R(\mathbf{s}_t, \mathbf{a}_t)$ with the above expression, and rearranging terms, we obtain

$$\delta_t^{\text{tot}} = \sum_{i \in \mathcal{N}} w_i^* \left( r_i(s_{i,t}, a_{i,t}) + \tfrac{\gamma}{w_i^*} V_\phi^{\text{tot}}(\mathbf{s}_{t+1}) - V_\phi^{\text{tot}}(\mathbf{s}_t) \right) + (1-\gamma)\,w^*.$$

Recognizing that the term inside parentheses corresponds exactly to $\delta_{i,t}^{\text{local}}$, we have

$$\delta_t^{\text{tot}} = \sum_{i \in \mathcal{N}} w_i^* \, \delta_{i,t}^{\text{local}} + (1-\gamma)\,w^*.$$

Applying generalized advantage estimation (GAE) recursively to both the global and local deltas establishes

$$\hat{A}_t^{\text{tot}} = \sum_{i \in \mathcal{N}} w_i^* \, \hat{A}_{i,t}^{\text{local}} + (1-\gamma)\,w^*,$$

which completes the proof. $\qquad\square$

### A.3 PROOF OF PROPOSITION 2

**Proposition 2.** *Assume that the joint policy factorizes as $\boldsymbol{\pi}_{\boldsymbol{\theta}}(\boldsymbol{a}_t \mid \boldsymbol{s}_t) = \prod_{i=1}^n \pi_{\theta_i}(a_{i,t} \mid s_{i,t})$, then the policy gradient of the joint policy can be expressed as a weighted sum of the local policy gradients: $\nabla_{\boldsymbol{\theta}} J(\boldsymbol{\pi}_{\boldsymbol{\theta}}) = \sum_{i=1}^n w_i^* \nabla_{\theta_i} J(\pi_{\theta_i})$, where $w_i^*$ are the mixing weights returned from the preference-based implicit reward learning.*

*Proof.* By policy factorization, $\log \boldsymbol{\pi}_{\boldsymbol{\theta}}(\mathbf{a}_t \mid \mathbf{s}_t) = \sum_{i=1}^{n} \log \pi_{\theta_i}(a_{i,t} \mid s_{i,t})$ and therefore $\nabla_{\boldsymbol{\theta}} \log \boldsymbol{\pi}_{\boldsymbol{\theta}}(\mathbf{a}_t \mid \mathbf{s}_t) = \left(\nabla_{\theta_1} \log \pi_{\theta_1}(a_t^1 \mid o_t^1), \ldots, \nabla_{\theta_n} \log \pi_{\theta_n}(a_t^n \mid o_t^n)\right)$. Using the factorization of $\hat{A}_t^{\text{tot}}$ and linearity of expectation,

$$\nabla_{\boldsymbol{\theta}} J(\boldsymbol{\pi}_{\boldsymbol{\theta}}) = \nabla_{\boldsymbol{\theta}} \mathbb{E}_t\Big[\log \boldsymbol{\pi}_{\boldsymbol{\theta}}(\mathbf{a}_t \mid \mathbf{s}_t) \big(\sum_{i=1}^{n} w_i^* \hat{A}_{i,t}^{\text{local}} + (1-\gamma) w^*\big)\Big]$$

$$= \sum_{i=1}^{n} w_i^* \mathbb{E}_t\Big[\nabla_{\boldsymbol{\theta}} \log \boldsymbol{\pi}_{\boldsymbol{\theta}}(\mathbf{a}_t \mid \mathbf{s}_t) \hat{A}_{i,t}^{\text{local}}\Big] + (1-\gamma) w^* \mathbb{E}_t\big[\nabla_{\boldsymbol{\theta}} \log \boldsymbol{\pi}_{\boldsymbol{\theta}}(\mathbf{a}_t \mid \mathbf{s}_t)\big].$$

Extracting the $i$-th block of the gradient and using that the $j \neq i$ blocks do not depend on $\theta_i$,

$$\mathbb{E}_t\Big[\nabla_{\boldsymbol{\theta}} \log \boldsymbol{\pi}_{\boldsymbol{\theta}}(\mathbf{a}_t \mid \mathbf{s}_t) \hat{A}_{i,t}^{\text{local}}\Big] = \Big(\underbrace{0,\ldots,0}_{j<i}, \mathbb{E}_t[\nabla_{\theta_i} \log \pi_{\theta_i}(a_{i,t} \mid s_{i,t}) \hat{A}_{i,t}^{\text{local}}], \underbrace{0,\ldots,0}_{j>i}\Big).$$

Hence, stacking across $i$ gives

$$\nabla_{\boldsymbol{\theta}} J(\boldsymbol{\pi}_{\boldsymbol{\theta}}) = \sum_{i=1}^{n} w_i^* \Big(\underbrace{0,\ldots,0}_{j<i}, \mathbb{E}_t[\nabla_{\theta_i} \log \pi_{\theta_i}(a_{i,t} \mid s_{i,t}) \hat{A}_{i,t}^{\text{local}}], \underbrace{0,\ldots,0}_{j>i}\Big)$$

$$+ (1-\gamma) w^* \mathbb{E}_t\big[\nabla_{\boldsymbol{\theta}} \log \boldsymbol{\pi}_{\boldsymbol{\theta}}(\mathbf{a}_t \mid \mathbf{s}_t)\big].$$

$$= \sum_{i=1}^{n} w_i^* \mathbb{E}_t[\nabla_{\theta_i} \log \pi_{\theta_i}(a_{i,t} \mid s_{i,t}) \hat{A}_{i,t}^{\text{local}}] + (1-\gamma) w^* \mathbb{E}_t\big[\nabla_{\boldsymbol{\theta}} \log \boldsymbol{\pi}_{\boldsymbol{\theta}}(\mathbf{a}_t \mid \mathbf{s}_t)\big].$$

By the definition of the local objective $J(\pi_{\theta_i})$, the $i$-th block is precisely $\nabla_{\theta_i} J(\pi_{\theta_i})$, proving the stated identity. Finally, the last term $\mathbb{E}_t[\nabla_{\boldsymbol{\theta}} \log \boldsymbol{\pi}_{\boldsymbol{\theta}}(\mathbf{a}_t \mid \mathbf{s}_t)]$ vanishes under on-policy sampling by the reparameterization-free (score-function) identity:

$$\mathbb{E}_t[\nabla_{\boldsymbol{\theta}} \log \boldsymbol{\pi}_{\boldsymbol{\theta}}(\mathbf{a}_t \mid \mathbf{s}_t)] = \mathbb{E}_{\mathbf{s}_t}\left[\sum_{\mathbf{a}_t} \nabla_{\boldsymbol{\theta}} \boldsymbol{\pi}_{\boldsymbol{\theta}}(\mathbf{a}_t \mid \mathbf{s}_t)\right] \tag{9}$$

$$= \mathbb{E}_{\mathbf{s}_t}\left[\nabla_{\boldsymbol{\theta}} \sum_{\mathbf{a}_t} \boldsymbol{\pi}_{\boldsymbol{\theta}}(\mathbf{a}_t \mid \mathbf{s}_t)\right] \tag{10}$$

$$= \mathbb{E}_{\mathbf{s}_t}[\nabla_{\boldsymbol{\theta}} 1] = \mathbf{0}, \tag{11}$$

since $\sum_{\mathbf{a}_t} \boldsymbol{\pi}_{\boldsymbol{\theta}}(\mathbf{a}_t \mid \mathbf{s}_t) = 1$ for all $\mathbf{s}_t$. Therefore, the bias term disappears, and the global policy gradient reduces to the weighted sum of local policy gradients. $\square$

# B EXPERIMENTAL DETAILS

This section provides a detailed overview of the experimental environments, the baseline algorithms used for comparison, and our proposed methods.

## B.1 ENVIRONMENTS

We evaluate our methods on two distinct multi-agent benchmarks: the discrete-action StarCraft II Multi-Agent Challenge (SMACv2) (Ellis et al., 2023) and the continuous-control Multi-Agent MuJoCo (MaMujoco) (de Witt et al., 2020).

### B.1.1 SMACV2

The StarCraft II Multi-Agent Challenge (SMACv2) (Ellis et al., 2023) is a challenging benchmark for cooperative MARL based on the popular real-time strategy game StarCraft II. It features a diverse set of micromanagement scenarios where a group of allied agents must defeat a group of enemy agents controlled by the game's built-in AI. SMACv2 improves upon its predecessor by introducing procedurally generated maps, which prevents agents from overfitting to a fixed environment layout and promotes the learning of more generalizable strategies. Agents have partially observable views

of the battlefield and must learn to coordinate their actions effectively. The action space is discrete, including moving, attacking specific enemies, and stopping.

For our experiments, we use a set of nine challenging symmetric and asymmetric scenarios across the three StarCraft II races:

- **Protoss**: 'protoss_5_vs_5', 'protoss_10_vs_10', 'protoss_10_vs_11'
- **Terran**: 'terran_5_vs_5', 'terran_10_vs_10', 'terran_10_vs_11'
- **Zerg**: 'zerg_5_vs_5', 'zerg_10_vs_10', 'zerg_10_vs_11'

The primary evaluation metric is the win rate, which is the percentage of episodes where the agents successfully defeat all enemy units. A sparse reward of $+1$ is given for winning an episode and $-1$ for losing, with no intermediate rewards.

### B.1.2 MAMUJOCO

Multi-Agent MuJoCo (MaMujoco) (de Witt et al., 2020) is a continuous-control benchmark for MARL, adapted from the popular single-agent MuJoCo environments. In these tasks, a single robotic morphology is decomposed into multiple agents, each controlling a subset of the robot's joints. The agents must learn to coordinate their continuous actions (joint torques) to achieve a common goal, such as locomotion. The state space is continuous and fully observable. We use the following four tasks:

- **Hopper-v2**: A two-legged robot where agents control different joints to achieve forward hopping.
- **Reacher-v2**: A robotic arm where agents control joints to reach a target location.
- **HalfCheetah-v2**: A two-legged cheetah-like robot where agents coordinate to run forward as fast as possible.
- **Swimmer-v2**: A snake-like robot where agents control joints to swim forward.

The evaluation metric is the total reward accumulated over a trajectory. The reward functions are dense and are based on task-specific objectives, such as forward velocity for Hopper and HalfCheetah. For our sparse-reward setting, we provide the final cumulative trajectory reward at the end of the episode. Specifically, instead of using per-step rewards, we provide agents with a single *trajectory-level* reward at the end of each episode. This reward is computed as the normalized cumulative return over the full trajectory, and preference labels are generated by comparing pairs of complete trajectories. This modification ensures that agents receive feedback only at the episode level, consistent with the sparse-reward formulation studied in this paper. All other environment dynamics, action spaces, and termination conditions remain unchanged.

### B.2 BASELINES

We compare our IMAP framework against the following baselines MARL. From this point onward, when describing the baselines and experimental settings, we use the notation $\mathbf{o}$ and $o_i$ instead of $\mathbf{s}$ and $s_i$ to reflect the practical setting where agents have access only to local observations rather than the full global state.

**SparseMAPPO**: This is the MAPPO algorithm (Yu et al., 2022) trained directly with sparse reward feedback. In SMACv2, the reward $r$ is $+1$ for a win and $-1$ for a loss. In MaMujoco, $R(\sigma)$ is the total cumulative reward of the trajectory $\sigma$, provided only at the final timestep. The MAPPO actor for each agent $i$ is updated by maximizing the clipped surrogate objective:

$$\mathcal{L}_{\text{CLIP}}(\theta_i) = \mathbb{E}_{\tau \sim \pi_{\theta_i}} \left[ \min \left( \rho_t(\theta_i) \hat{A}_t^{\text{tot}}, \text{clip}(\rho_t(\theta_i), 1 - \epsilon, 1 + \epsilon) \hat{A}_t^{\text{tot}} \right) \right]$$

where $\rho_t(\theta_i) = \frac{\pi_{\theta_i}(a_{t,i}|o_{i,t})}{\pi_{\theta_k}(a_{t,i}|o_{i,t})}$ is the importance sampling ratio, and $\hat{A}_t^{\text{tot}}$ is the global advantage estimate from a centralized critic trained on the sparse global rewards.

**SparseHAPPO**: This is based on the HAPPO algorithm (Kuba et al., 2021) trained directly with sparse reward feedback. HAPPO (Heterogeneous-Agent Proximal Policy Optimization) extends

MAPPO to handle heterogeneous agents with different observation and action spaces. Similar to SparseMAPPO, it uses the sparse trajectory-level rewards to compute advantages. The key difference is that HAPPO uses sequential policy updates with trust region constraints to ensure monotonic improvement for each agent. Each agent $i$'s policy is updated using:

$$\mathcal{L}_{\text{HAPPO}}(\theta_i) = \mathbb{E}_{\tau \sim \pi_{\theta_i}} \left[ \min \left( \rho_t(\theta_i) \hat{A}_t^i, \text{clip}(\rho_t(\theta_i), 1 - \epsilon, 1 + \epsilon) \hat{A}_t^i \right) \right]$$

where $\hat{A}_t^i$ is computed using individual value baselines that account for the sequential update structure.

**Online-IPL**: This baseline adapts the concept of IPL (Bui et al., 2025a; Hejna & Sadigh, 2024) to the online MARL setting, where the global value functions $\widetilde{Q}_{\text{tot}}$ and $\widetilde{V}_{\text{tot}}$ are used to extract local policies. In IPL, an implicit reward function is inferred from a dataset of preferences over pairs of trajectories $(\sigma_1, \sigma_2)$. The preference probability is typically modeled using the Bradley–Terry (BT) model (Bradley & Terry, 1952):

$$P(\sigma_1 \succ \sigma_2) = \frac{\exp\left(\sum_{t=0}^{T} \gamma^t \hat{R}(\mathbf{s}_t^1, \mathbf{a}_t^1)\right)}{\exp\left(\sum_{t=0}^{T} \gamma^t \hat{R}(\mathbf{s}_t^1, \mathbf{a}_t^1)\right) + \exp\left(\sum_{t=0}^{T} \gamma^t \hat{R}(\mathbf{s}_t^2, \mathbf{a}_t^2)\right)},$$

where the implicit reward is defined as

$$\hat{R}(\mathbf{s}, \mathbf{a}) = \widetilde{Q}_{\text{tot}}(\mathbf{s}, \mathbf{a}) - \gamma \mathbb{E}_{\mathbf{s}'}[\widetilde{V}_{\text{tot}}(\mathbf{s}')].$$

At each training step, preferences are generated from sparse environment outcomes (e.g., a winning trajectory is preferred over a losing one). The global functions $\widetilde{Q}_{\text{tot}}$ and $\widetilde{V}_{\text{tot}}$ are updated (via decomposition into local value functions with a mixing network), and local policies are extracted using behavior cloning with importance weighting (Bui et al., 2025b;a; Wang et al., 2022):

$$\max_{\pi_i} \ \mathbb{E}_{(s_i, a_i) \sim \mathcal{D}}\left[ \log \pi_i(a_i \mid o_i) \ \exp\left(\frac{\widetilde{Q}_{\text{tot}}(\mathbf{s}, \mathbf{a}) - \widetilde{V}_{\text{tot}}(\mathbf{s})}{\beta}\right) \right],$$

where $\beta$ is a temperature parameter and $\mathcal{D}$ denotes the dataset of state–action pairs induced by the global policy extracted from $Q_{\text{tot}}$. The updated policies are then used to sample additional trajectories, which are added to the replay buffer to refine preference learning and further update $Q_{\text{tot}}$ and $V_{\text{tot}}$. This process is repeated until convergence.

**SL-MAPPO**: This baseline directly augments MAPPO with a conventional supervised learning-based reward model trained from preference data (Christiano et al., 2017). Specifically, a neural network parameterized by $\psi$ is trained to predict which of two trajectories is preferred. Given a dataset of preferences $\mathcal{D} = \{(\sigma_1^{(j)}, \sigma_2^{(j)}, y^{(j)})\}_{j=1}^{M}$, where $y^{(j)} \in \{0, 1\}$ indicates whether $\sigma_1^{(j)}$ or $\sigma_2^{(j)}$ is preferred, the reward model $\hat{R}_\psi$ is optimized using the standard Bradley–Terry loss:

$$\mathcal{L}_{\text{SL}}(\psi) = -\mathbb{E}_{(\sigma_1, \sigma_2, y) \sim \mathcal{D}}\left[ y \log P_{\hat{R}_\psi}(\sigma_1 \succ \sigma_2) + (1 - y) \log P_{\hat{R}_\psi}(\sigma_2 \succ \sigma_1) \right],$$

where $P_{\hat{R}_\psi}(\sigma_1 \succ \sigma_2)$ denotes the probability, induced by the learned reward model, that trajectory $\sigma_1$ is preferred over $\sigma_2$. Once trained, $\hat{R}_\psi(\mathbf{s}_t, \mathbf{a}_t)$ provides dense, transition-level reward estimates, which are then used as global rewards to train the MAPPO agents in place of the sparse environment signals. Similar to our IMAP algorithm, this also converts high-level preference supervision into stepwise feedback, enabling more efficient policy learning under sparse reward settings.

### B.3    IMAP: INVERSE PREFERENCE-GUIDED MULTI-AGENT POLICY OPTIMIZATION

This section provides a detailed description of our **IMAP** framework, which is designed for online multi-agent reinforcement learning in environments with sparse rewards. IMAP leverages online inverse preference learning to recover a dense, implicit reward signal, which then guides policy optimization using a PPO-style algorithm. We present two variants of IMAP based on the source of preference labels: a rule-based approach and an LLM-based approach.

### B.3.1 CORE FRAMEWORK

Let a trajectory $\sigma = \{(\mathbf{s}_0, \mathbf{a}_0), \ldots, (\mathbf{s}_T, \mathbf{a}_T)\}$ be a sequence of state-action pairs, and let $R(\sigma)$ be the final sparse reward for that trajectory. Given two trajectories, $\sigma_1$ and $\sigma_2$, collected from the environment, a preference is established based on a simple rule: $\sigma_1 \succ \sigma_2$ (read as $\sigma_1$ is preferred over $\sigma_2$) if $R(\sigma_1) > R(\sigma_2)$.

The core of IMAP is to learn an implicit, dense reward function $R(\mathbf{s}_t, \mathbf{a}_t)$ that explains these trajectory-level preferences. Following the inverse preference learning (IPL) framework, we operate in the Q-function space. The implicit reward is defined via the inverse soft Bellman operator:

$$R(\mathbf{s}_t, \mathbf{a}_t) = (\mathcal{T}^* Q_{\text{tot}})(\mathbf{s}_t, \mathbf{a}_t) = Q_{\text{tot}}(\mathbf{s}_t, \mathbf{a}_t) - \gamma \mathbb{E}_{\mathbf{s}_{t+1} \sim P(\cdot | \mathbf{s}_t, \mathbf{a}_t)}[V_{\text{tot}}(\mathbf{s}_{t+1})] \quad (12)$$

where $V_{\text{tot}}(\mathbf{s}) = \beta \log \sum_{\mathbf{a}} \exp(Q_{\text{tot}}(\mathbf{s}, \mathbf{a})/\beta)$ is the soft value function.

The probability of preferring $\sigma_1$ over $\sigma_2$ is modeled using the Bradley-Terry model (Bradley & Terry, 1952):

$$P(\sigma_1 \succ \sigma_2 | Q_{\text{tot}}) = \frac{\exp\left(\sum_{t=0}^{T} \gamma^t (\mathcal{T}^* Q_{\text{tot}})(\mathbf{s}_t^1, \mathbf{a}_t^1)\right)}{\exp\left(\sum_{t=0}^{T} \gamma^t (\mathcal{T}^* Q_{\text{tot}})(\mathbf{s}_t^1, \mathbf{a}_t^1)\right) + \exp\left(\sum_{t=0}^{T} \gamma^t (\mathcal{T}^* Q_{\text{tot}})(\mathbf{s}_t^2, \mathbf{a}_t^2)\right)} \quad (13)$$

The Q-function is learned by maximizing the log-likelihood of the preference data $\mathcal{P}$ collected online, with an additional regularization term $\phi(\cdot)$ to prevent reward scaling issues:

$$\max_{Q_{\text{tot}}} \mathcal{L}(Q_{\text{tot}} | \mathcal{P}) = \sum_{(\sigma_1, \sigma_2) \in \mathcal{P}} \left[ \log P(\sigma_1 \succ \sigma_2 | Q_{\text{tot}}) + \sum_{t \in \sigma_1 \cup \sigma_2} \phi((\mathcal{T}^* Q_{\text{tot}})(\mathbf{s}_t, \mathbf{a}_t)) \right] \quad (14)$$

To handle the large state-action space in MARL, we use value decomposition. The global Q- and V-functions are factorized using a linear mixing network $\mathcal{M}_w$:

$$Q_{\text{tot}}(\mathbf{s}, \mathbf{a}) = \mathcal{M}_w[\mathbf{q}(\mathbf{s}, \mathbf{a})] = \sum_{i=1}^{n} w_i q_i(s_i, a_i) + w \quad (15)$$

$$V_{\text{tot}}(\mathbf{s}) = \mathcal{M}_w[\mathbf{v}(\mathbf{s})] = \sum_{i=1}^{n} w_i v_i(s_i) + w \quad (16)$$

where $\mathbf{q} = \{q_1, \ldots, q_n\}$ and $\mathbf{v} = \{v_1, \ldots, v_n\}$ denote the local value functions. For consistency, we employ the same mixing network—a linear combination with shared coefficients $w_i$—to factorize both $Q_{\text{tot}}$ and $V_{\text{tot}}$.

The relation between $Q_{\text{tot}}$ and $V_{\text{tot}}$ is maintained by minimizing the *Extreme-V* loss objective (Garg et al., 2023):

$$\mathcal{J}(\mathbf{v} \mid \mathbf{q}) = \mathbb{E}_{(\mathbf{s}, \mathbf{a}) \sim \mathcal{D}} \left[ \exp\left( \frac{\mathcal{M}_w[\mathbf{q}(\mathbf{s}, \mathbf{a})] - \mathcal{M}_w[\mathbf{v}(\mathbf{s})]}{\beta} \right) \right] - \mathbb{E}_{(\mathbf{s}, \mathbf{a}) \sim \mathcal{D}} \left[ \frac{\mathcal{M}_w[\mathbf{q}(\mathbf{s}, \mathbf{a})] - \mathcal{M}_w[\mathbf{v}(\mathbf{s})]}{\beta} \right] - 1, \quad (17)$$

where $\mathcal{D}$ is the buffer of collected transitions. Following Garg et al. (2023), updating $\mathbf{v}$ by minimizing $\mathcal{J}(\mathbf{v} \mid \mathbf{q})$ guarantees that $V_{\text{tot}}$ converges to the *log-sum-exp* of $Q_{\text{tot}}$, i.e.,

$$V_{\text{tot}}(\mathbf{s}) \;\rightarrow\; \beta \log \sum_{\mathbf{a}} \exp\left( \frac{Q_{\text{tot}}(\mathbf{s}, \mathbf{a})}{\beta} \right),$$

which yields a smooth and consistent approximation of the maximum operator, thereby facilitating stable training, particularly in environments with continuous action spaces such as MAMuJoCo (de Witt et al., 2020).

The online learning procedure is summarized in Algorithm 1, where the notation $\mathbf{o}$ is used in place of $\mathbf{s}$ to emphasize that agents have access only to partial observations (both local and global) rather than the full environment states.

In the early stages of training, stochastic policies often produce trajectories with similar cumulative returns, providing limited or noisy preference information. To address this cold-start issue, IMAP

adopts a two-stage strategy. First, we initialize the preference buffer with a small set of randomly sampled trajectories to ensure sufficient diversity in early feedback. Second, we employ the rule-based preference model (IMAP-Rule) to generate coarse but informative preferences before transitioning to the learned or LLM-guided models as the policy improves. This hybrid initialization effectively bootstraps the preference-learning process, promoting early exploration and ensuring that preference comparisons become increasingly meaningful as training progresses.

---

**Algorithm 1 IMAP: Implicit Multi-agent Preference learning**

---

1: **Initialize:** Actor networks $\pi_{\theta_i}$ for each agent $i$, critic network $V_\phi$, local Q-networks $q_{\psi_i}$, local V-networks $v_{\xi_i}$, mixing network $\mathcal{M}_w$.
2: **Initialize:** Experience buffer $\mathcal{D}$ and Preference buffer $\mathcal{P}$.
3: **for** each training iteration **do**
4:     Collect a batch of trajectories $\{\sigma_k\}$ by executing policies $\{\pi_{\theta_i}\}$ in the environment. Store transitions in $\mathcal{D}$.
5:     **// Generate Preferences**
6:     Sample pairs of trajectories $(\sigma_i, \sigma_j)$ from the collected batch.
7:     **if** $\sigma_i \succ \sigma_j$ **then**
8:         Add $(\sigma_i \succ \sigma_j)$ to preference buffer $\mathcal{P}$.
9:     **else if** $\sigma_j \succ \sigma_i$ **then**
10:        Add $(\sigma_j \succ \sigma_i)$ to preference buffer $\mathcal{P}$.
11:     **end if**
12:     **// Update Implicit Reward Model**
13:     **for** several gradient steps **do**
14:         Sample a mini-batch of preferences from $\mathcal{P}$.
15:         Update local Q-networks $\{\psi_i\}$ and mixer $w$ by maximizing the preference log-likelihood $\mathcal{L}(Q_{\text{tot}}|\mathcal{P})$.
16:         Update local V-networks $\{\xi_i\}$ by minimizing the extreme-V loss $\mathcal{J}(\mathbf{v}|\mathbf{q})$.
17:     **end for**
18:     **// Update Policies (PPO)**
19:     **for** each PPO epoch **do**
20:         For each transition $(\mathbf{o}_t, \mathbf{a}_t, \mathbf{o}_{t+1})$ in $\mathcal{D}$:
21:         Compute implicit rewards:

$$R_t(\mathbf{o}_t, \mathbf{a}_t) = \mathcal{M}_w[\mathbf{q}(\mathbf{o}_t, \mathbf{a}_t)] - \gamma \mathcal{M}_w[\mathbf{v}(\mathbf{o}_{t+1})]$$
$$r_i(o_{i,t}, a_{i,t}) = q_i(o_{i,t}, a_{i,t}) - \gamma \, v_i(o_{i,t+1}).$$

22:         Compute global and local advantage estimates $\hat{A}_t^{\text{tot}}$ and $\hat{A}_{i,t}^{\text{local}}$ using GAE with $R_t$ and $r_i$.
23:         Update actor networks $\{\theta_i\}$ using the PPO clipped surrogate objective with $\hat{A}_{i,t}^{\text{local}}$.
24:         Update centralized critic $V_\phi$ by minimizing the value loss against the implicit returns.
25:     **end for**
26: **end for**

---

### B.3.2 LLM-BASED PREFERENCE GENERATION

We describe our IMAP that uses LLM to generate preference feedback. This variant enhances the preference generation process by replacing the simple rule-based comparison with a sophisticated LLM. The LLM is prompted with detailed, context-rich descriptions of two trajectories and is asked to provide a preference judgment. This allows for more nuanced and semantically meaningful supervision that can capture aspects of strategy beyond the final score.

Instead of relying solely on the sparse reward, we extract a rich set of features from each trajectory to form a natural language prompt. These features include terminal state information (e.g., remaining health of allies and enemies, number of deaths) and trajectory statistics (e.g., total steps). The LLM's task is to evaluate these summaries and determine which trajectory demonstrates superior performance based on high-level strategic objectives.

An example prompt used to query the LLM for preference labels in SMAC scenarios is shown below. This template is designed to provide sufficient context for the LLM to make an informed decision.

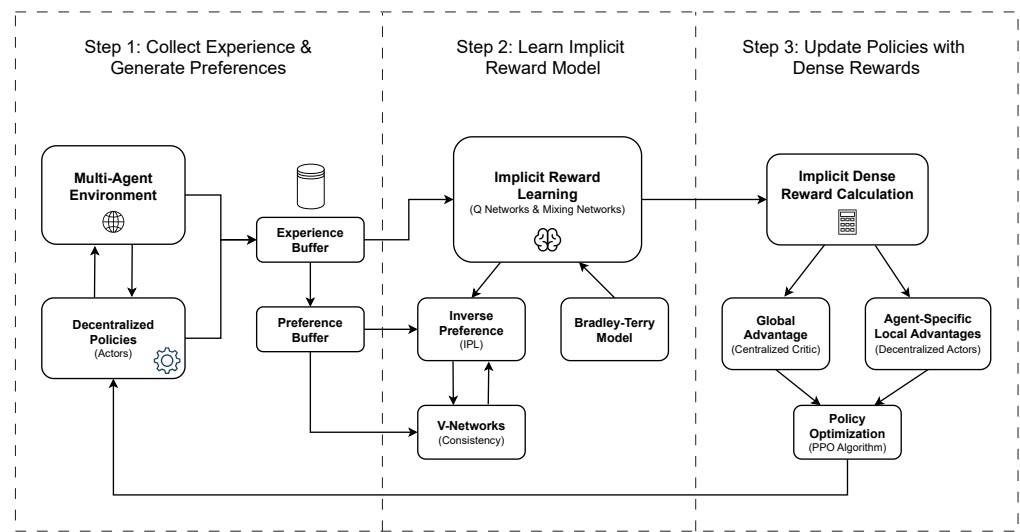

Figure 2: IMAP Workflow Diagram

## C  IMPLEMENTATION DETAILS

All experiments were conducted using 5 random seeds for each algorithm-environment pair to ensure statistical significance. The network architecture for agents in SMACv2 consists of a Gated Recurrent Unit (GRU) with a 128-dimensional hidden state to process the history of observations, followed by two fully-connected layers. For MaMujoco, a standard Multi-Layer Perceptron (MLP) with two hidden layers of size 256 was used for both actors and critics. Key hyperparameters for our IMAP framework and the baselines are detailed in Table 5.

Table 5: Key hyperparameters used for experiments.

| Hyperparameter | SMACv2 | MaMujoco |
|---|---|---|
| Optimizer | Adam | Adam |
| Learning Rate (Actor) | $5 \times 10^{-4}$ | $3 \times 10^{-4}$ |
| Learning Rate (Critic) | $5 \times 10^{-4}$ | $3 \times 10^{-4}$ |
| Discount Factor ($\gamma$) | 0.99 | 0.99 |
| PPO Clipping ($\epsilon$) | 0.2 | 0.2 |
| GAE Lambda ($\lambda$) | 0.95 | 0.95 |
| Number of PPO Epochs | 10 | 10 |
| Minibatch Size | 256 | 512 |
| Entropy Coefficient | 0.01 | 0.01 |
| GRU Hidden Size | 128 | N/A |
| MLP Hidden Size | N/A | 256 |

### C.1  EXPERIMENTAL RESULTS

This appendix provides a detailed breakdown of the empirical results for the main experiments and ablation studies. All reported values are the mean of five independent runs with different random seeds, accompanied by the standard deviation.

#### C.1.1  MAIN EXPERIMENT RESULTS

In this section, we present the primary comparison of our IMAP framework against established baselines in sparse-reward multi-agent reinforcement learning. We evaluate **SparseMAPPO** and

Table 4: Sample prompt to generate preference data in SMAC environments.

---

**Prompt**

```
You are a helpful and honest judge of good game playing and progress in the StarCraft
Multi-Agent Challenge game. Always answer as helpfully as possible, while being
truthful.
If you don't know the answer to a question, please don't share false information.
I'm looking to have you evaluate a scenario in the StarCraft Multi-Agent Challenge.
Your role will be to assess how much the actions taken by multiple agents in a given
situation have contributed to achieving victory.

The basic information for the evaluation is as follows.

- Scenario : 5m_vs_6m
- Allied Team Agent Configuration : five Marines(Marines are ranged units in StarCraft
2).
- Enemy Team Agent Configuration : six Marines(Marines are ranged units in StarCraft
2).
- Situation Description : The situation involves the allied team and the enemy team
engaging in combat, where victory is achieved by defeating all the enemies.
- Objective : Defeat all enemy agents while ensuring as many allied agents as possible
survive.
* Important Notice : You should prefer the trajectory where our allies' health is
preserved while significantly reducing the enemy's health. In similar situations, you
should prefer shorter trajectory lengths.

I will provide you with two trajectories, and you should select the better trajectory
based on the outcomes of these trajectories. Regarding the trajectory, it will inform
you about the final states, and you should select the better case based on these two
trajectories.

[Trajectory 1]
1. Final State Information
    1) Allied Agents Health : 0.000, 0.000, 0.067, 0.067, 0.000
    2) Enemy Agents Health : 0.000, 0.000, 0.000, 0.000, 0.000, 0.040
    3) Number of Allied Deaths : 3
    4) Number of Enemy Deaths : 5
    5) Total Remaining Health of Allies : 0.133
    6) Total Remaining Health of Enemies : 0.040
2. Total Number of Steps : 28

[Trajectory 2]
1. Final State Information
    1) Allied Agents Health : 0.000, 0.000, 0.000, 0.000, 0.000
    2) Enemy Agents Health : 0.120, 0.000, 0.000, 0.000, 0.000, 0.200
    3) Number of Allied Deaths : 5
    4) Number of Enemy Deaths : 4
    5) Total Remaining Health of Allies : 0.000
    6) Total Remaining Health of Enemies : 0.320
2. Total Number of Steps : 23

Your task is to inform which one is better between [Trajectory1] and [Trajectory2]
based on the information mentioned above. For example, if [Trajectory 1] seems better,
output #1, and if [Trajectory 2] seems better, output #2. If it's difficult to judge or
they seem similar, please output #0.
* Important : Generally, it is considered better when fewer allied agents are killed or
injured while inflicting more damage on the enemy.

Omit detailed explanations and just provide the answer.
```

---

**SparseHAPPO**, which use the raw sparse reward; **Online-IPL** and **SL-MAPPO**, which learn explicit reward models; **IMAP-Rule**, our method using final outcomes for preference; and **IMAP-LLM**, our method leveraging Qwen-4B for nuanced preference generation.

**SMACv2.** The performance on the StarCraft II Multi-Agent Challenge (SMACv2) (Ellis et al., 2023) is measured by the mean win rate (%). The results across all three races - Protoss, Terran, and Zerg - are detailed in Tables 6, 7, and 8.

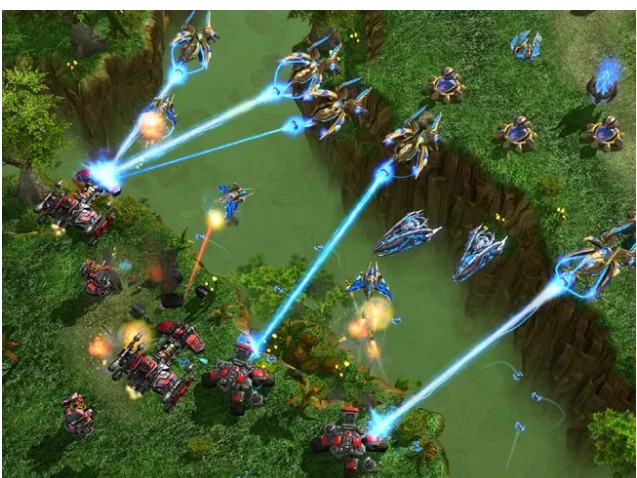

Figure 3: StarCraft II gameplay.

Table 6: Mean win rate (%) $\pm$ standard deviation on SMACv2 Protoss scenarios. Our IMAP framework, especially the LLM-enhanced version, demonstrates a substantial performance improvement over all baselines, highlighting its effectiveness in overcoming sparse rewards.

| | Protoss Scenarios | | |
| --- | --- | --- | --- |
| **Algorithm** | protoss_5_vs_5 | protoss_10_vs_10 | protoss_10_vs_11 |
| SparseMAPPO | $0.00 \pm 0.00$ | $0.00 \pm 0.00$ | $0.00 \pm 0.00$ |
| SparseHAPPO | $11.61 \pm 3.03$ | $0.00 \pm 0.00$ | $0.00 \pm 0.00$ |
| Online-IPL | $15.03 \pm 3.80$ | $9.35 \pm 2.50$ | $7.54 \pm 2.33$ |
| SL-MAPPO | $26.97 \pm 6.00$ | $21.38 \pm 3.58$ | $15.67 \pm 3.48$ |
| IMAP-Rule | $44.00 \pm 3.16$ | $33.64 \pm 2.78$ | $23.08 \pm 3.99$ |
| IMAP-LLM | $\mathbf{48.34 \pm 2.76}$ | $\mathbf{37.76 \pm 2.30}$ | $\mathbf{25.24 \pm 4.69}$ |

In the Protoss scenarios, standard SparseMAPPO completely fails to learn, achieving a 0% win rate across all maps. SparseHAPPO shows some improvement in the 5_vs_5 scenario (11.61%), demonstrating that the heterogeneous-agent PPO approach can provide marginal benefits, but still fails completely in the larger 10_vs_10 and 10_vs_11 maps. This underscores the critical need for an effective reward shaping mechanism in sparse settings. While baselines that learn an explicit reward model (Online-IPL and SL-MAPPO) show some progress, they still struggle to achieve high performance. In stark contrast, our IMAP framework provides a significant leap in performance. The rule-based IMAP already achieves strong results, validating our core approach of learning an *implicit* reward signal in the Q-space. The LLM-based variant further pushes this boundary, indicating that the semantically rich preferences from Qwen-4B provide a superior and more nuanced learning signal for complex coordination tasks.

Table 7: Mean win rate (%) $\pm$ standard deviation on SMACv2 Terran scenarios. The results underscore the robustness of the IMAP framework, which consistently delivers state-of-the-art performance across different unit compositions and difficulties.

| Algorithm | Terran Scenarios | | |
| --- | --- | --- | --- |
| | terran_5_vs_5 | terran_10_vs_10 | terran_10_vs_11 |
| SparseMAPPO | $31.96 \pm 3.26$ | $23.12 \pm 4.24$ | $0.00 \pm 0.00$ |
| SparseHAPPO | $35.46 \pm 3.58$ | $23.47 \pm 2.85$ | $0.00 \pm 0.00$ |
| Online-IPL | $32.47 \pm 4.42$ | $23.40 \pm 2.66$ | $10.65 \pm 1.78$ |
| SL-MAPPO | $38.11 \pm 5.57$ | $28.22 \pm 4.16$ | $14.72 \pm 2.45$ |
| IMAP-Rule | $46.84 \pm 2.93$ | $38.18 \pm 2.82$ | $19.71 \pm 2.08$ |
| IMAP-LLM | $\mathbf{55.53 \pm 3.11}$ | $\mathbf{38.12 \pm 3.06}$ | $\mathbf{26.30 \pm 4.59}$ |

The performance trends continue in the Terran maps. SparseHAPPO achieves modest improvements over SparseMAPPO in the 5_vs_5 (35.46% vs 31.96%) and 10_vs_10 scenarios, but both methods still struggle with the challenging 10_vs_11 asymmetric scenario. IMAP-LLM achieves a remarkable **55.53% win rate** in 'terran_5_vs_5', significantly outperforming the next best baseline (SL-MAPPO) by over 17 percentage points. This widening performance gap suggests that as scenario complexity increases, the quality of the preference signal becomes paramount. LLM-guided preferences prove more effective at capturing the strategic nuances of success—such as minimizing damage or efficient target-firing—that simple win/loss signals cannot convey.

Table 8: Mean win rate (%) $\pm$ standard deviation on SMACv2 Zerg scenarios. The consistent superiority of IMAP highlights the general applicability of our implicit reward learning framework across diverse multi-agent challenges.

| Algorithm | Zerg Scenarios | | |
| --- | --- | --- | --- |
| | zerg_5_vs_5 | zerg_10_vs_10 | zerg_10_vs_11 |
| SparseMAPPO | $18.45 \pm 3.99$ | $10.40 \pm 2.54$ | $4.63 \pm 1.63$ |
| SparseHAPPO | $23.01 \pm 3.92$ | $13.26 \pm 2.02$ | $4.76 \pm 1.60$ |
| Online-IPL | $28.23 \pm 3.46$ | $14.40 \pm 2.26$ | $4.78 \pm 1.91$ |
| SL-MAPPO | $25.80 \pm 3.74$ | $20.98 \pm 1.37$ | $8.73 \pm 2.72$ |
| IMAP-Rule | $42.14 \pm 3.44$ | $27.78 \pm 3.65$ | $17.24 \pm 1.77$ |
| IMAP-LLM | $\mathbf{43.82 \pm 5.12}$ | $\mathbf{29.75 \pm 2.75}$ | $\mathbf{18.39 \pm 2.22}$ |

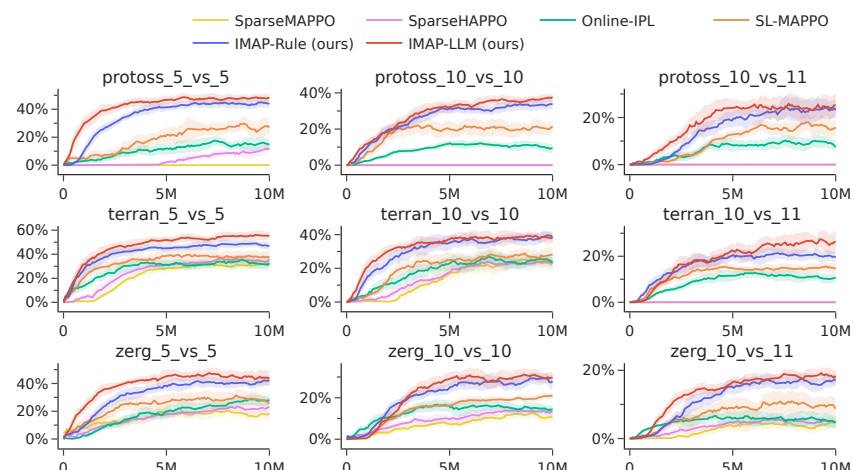

Figure 4: Learning curves on nine SMACv2 scenarios. The mean win rate is plotted against millions of timesteps. Our IMAP methods (IMAP-Rule and IMAP-LLM) consistently outperform the baseline algorithms (SparseMAPPO, Online-IPL, and SL-MAPPO) across all Protoss, Terran, and Zerg maps, demonstrating superior sample efficiency and final performance.

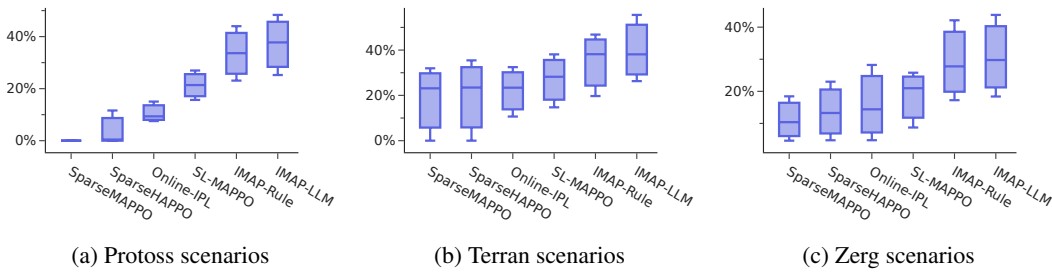

|          (a) Protoss scenarios          |          (b) Terran scenarios          |          (c) Zerg scenarios          |

Figure 5: Box plots summarizing the final win rate distributions of all algorithms on the SMACv2 scenarios. The plots clearly show that the IMAP variants achieve significantly higher and more stable performance compared to all baselines across all three races.

**MaMujoco.** For the Multi-Agent MuJoCo (MaMujoco) continuous control tasks (de Witt et al., 2020), performance is measured by the mean total episode reward. Results are presented in Table 9. Note that LLM-based variants are not applied in this setting, owing to the difficulty of constructing meaningful textual prompts from purely numerical, non-interpretable features.

Table 9: Mean total reward $\pm$ standard deviation on MaMujoco tasks. IMAP-Rule consistently outperforms all baselines, demonstrating its effectiveness in continuous control domains with sparse rewards.

| Algorithm | Hopper-v2 | Reacher-v2 | HalfCheetah-v2 | Swimmer-v2 |
|---|---|---|---|---|
| SparseMAPPO | $188.57 \pm 18.34$ | $-488.88 \pm 76.07$ | $1824.33 \pm 181.46$ | $16.29 \pm 2.12$ |
| SparseHAPPO | $215.41 \pm 30.01$ | $-497.58 \pm 77.65$ | $1926.69 \pm 224.82$ | $20.54 \pm 2.89$ |
| Online-IPL | $243.51 \pm 9.67$ | $-359.19 \pm 17.31$ | $1777.58 \pm 341.44$ | $23.10 \pm 0.81$ |
| SL-MAPPO | $296.45 \pm 19.09$ | $-358.15 \pm 15.53$ | $2479.37 \pm 337.53$ | $29.84 \pm 1.21$ |
| IMAP | $\mathbf{361.22 \pm 16.19}$ | $\mathbf{-182.10 \pm 13.25}$ | $\mathbf{3853.58 \pm 359.02}$ | $\mathbf{33.98 \pm 1.04}$ |

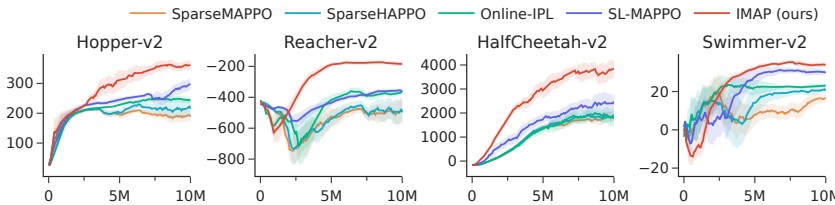

Figure 6: Learning curves on four continuous control tasks from the MaMujoco benchmark. The mean total reward is plotted over 10 million timesteps. Our IMAP framework (red) achieves substantially higher rewards than all baselines, highlighting its effectiveness in sparse-reward continuous control settings.

In MaMujoco, IMAP-Rule achieves substantially better performance than all baselines across every task. For instance, in 'HalfCheetah-v2', it scores **over 1300 points higher** than the best baseline, SL-MAPPO, and more than doubles the score of SparseMAPPO. Similarly, it drastically improves the negative reward in the challenging 'Reacher-v2' task. This demonstrates that our core implicit reward learning mechanism is highly general, adapting well to continuous action spaces and successfully translating sparse trajectory-level outcomes into dense, actionable reward signals without requiring LLM guidance.

## C.2 ABLATION 1: DUAL VS. SINGLE ADVANTAGE STREAMS

This ablation study investigates the impact of our dual advantage stream architecture. We compare **IMAP-LA** (Local Advantage, our full model) against **IMAP-GA** (Global Advantage), which uses a single, shared advantage stream for all agents. This comparison isolates the benefit of providing agent-specific learning signals.

Table 10: Ablation on advantage streams in SMACv2. Comparing IMAP-LA (dual stream) with IMAP-GA (single stream). The use of local, agent-specific advantages (IMAP-LA) consistently yields better performance, especially in more complex scenarios.

| | Protoss | | Terran | | Zerg | |
|---|---|---|---|---|---|---|
| **Scenario** | IMAP-GA | IMAP-LA | IMAP-GA | IMAP-LA | IMAP-GA | IMAP-LA |
| 5_vs_5 | $43.52 \pm 2.21$ | **44.00 ± 3.16** | $45.26 \pm 2.15$ | **46.84 ± 2.93** | $38.48 \pm 5.48$ | **42.14 ± 3.44** |
| 10_vs_10 | $31.71 \pm 2.19$ | **33.64 ± 2.78** | $36.23 \pm 3.26$ | **38.18 ± 2.82** | $26.15 \pm 1.51$ | **27.78 ± 3.65** |
| 10_vs_11 | $22.01 \pm 3.75$ | **23.08 ± 3.99** | $17.46 \pm 4.25$ | **19.71 ± 2.08** | $14.42 \pm 2.80$ | **17.24 ± 1.77** |

Table 11: Ablation on advantage streams in MaMujoco. The benefit of dual advantage streams is even more pronounced in continuous control, where precise credit assignment is critical for coordinated motion.

| **Environment** | IMAP-GA (Single Stream) | IMAP-LA (Dual Stream) |
|---|---|---|
| Hopper-v2 | $296.48 \pm 27.55$ | **361.22 ± 16.19** |
| Reacher-v2 | $-316.83 \pm 67.58$ | **-182.10 ± 13.25** |
| HalfCheetah-v2 | $3328.71 \pm 285.84$ | **3853.58 ± 359.02** |
| Swimmer-v2 | $31.89 \pm 1.18$ | **33.98 ± 1.04** |

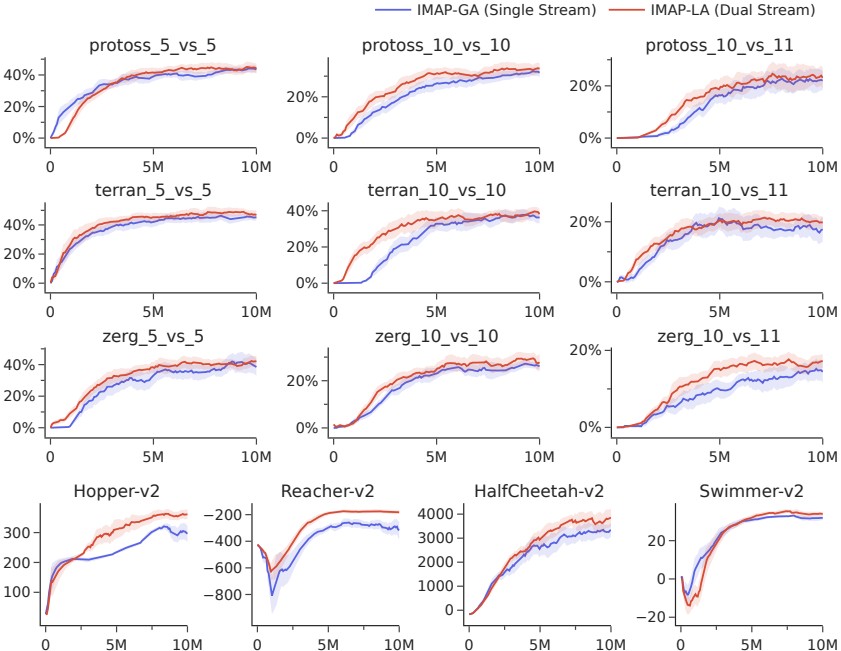

Figure 7: Ablation study comparing the performance of IMAP with a single global advantage stream (IMAP-GA) versus our proposed dual advantage stream (IMAP-LA). The top row displays results on SMACv2, and the bottom row shows results on MaMujoco. The dual-stream architecture consistently leads to better performance, confirming the benefit of agent-specific credit assignment.

Across both SMACv2 and MaMujoco, IMAP-LA consistently outperforms IMAP-GA. The performance gap is particularly noticeable in MaMujoco tasks like 'Hopper-v2' and 'Reacher-v2', where the dual-stream model shows a dramatic improvement. This confirms our hypothesis that providing differentiated, agent-specific credit through a local advantage stream is crucial for effective multi-agent coordination. A single global advantage stream, while still effective, can obscure individual contributions and assign credit inaccurately, leading to less efficient and stable policy updates, especially in tasks requiring fine-grained, heterogeneous actions.

## C.3 ABLATION 2: COMPARISON OF DIFFERENT LLM MODELS

This study compares the performance of IMAP when guided by different LLMs for preference annotation. We tested three models of varying sizes: **Gemma3-270m**, **Gemma3-4B**, and **Qwen-4B**. Larger models such as ChatGPT or Gemini were not considered due to the prohibitive cost of generating feedback in an online setting. Due to the computational cost of inference, this comparison was conducted on the challenging '5_vs_5' scenarios from SMACv2.

Table 12: Comparison of different LLMs for preference generation in IMAP on SMACv2 '5_vs_5' scenarios. The larger and more capable Qwen-4B model provides the best guidance, leading to the highest win rates.

| | **IMAP with Different LLMs** | | |
|---|---|---|---|
| Scenario | Gemma3-270m | Gemma3-4B | Qwen-4B |
| protoss_5_vs_5 | $30.95 \pm 5.45$ | $49.08 \pm 3.13$ | **$48.34 \pm 2.76$** |
| terran_5_vs_5 | $32.71 \pm 4.60$ | $51.71 \pm 3.17$ | **$55.53 \pm 3.11$** |
| zerg_5_vs_5 | $27.83 \pm 5.15$ | $42.74 \pm 3.51$ | **$43.82 \pm 5.12$** |

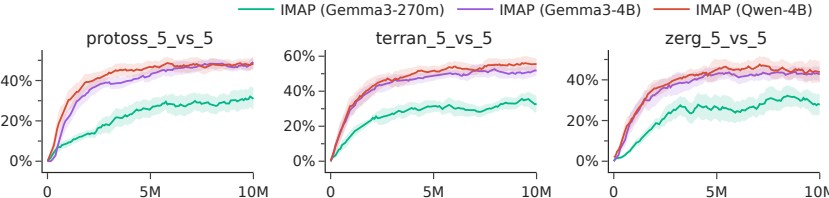

Figure 8: Performance comparison of the IMAP framework using different Large Language Models for preference generation on SMACv2 5_vs_5 scenarios. The results show a clear trend where larger, more capable models (Gemma3-4B and Qwen-4B) provide more effective guidance, leading to higher win rates than the smaller Gemma3-270m model.

The results clearly show a strong correlation between the scale of the LLM and the final performance of the trained agents. While even the small Gemma3-270m model provides a learning signal superior to the baselines, the larger Gemma3-4B and Qwen-4B models achieve progressively better results. Qwen-4B, the most capable model in our test set, consistently leads to the highest win rates. This is a key finding: the quality and nuance of the preference labels are directly influenced by the language model's reasoning capabilities. More advanced models can better interpret complex state information and identify subtle strategic advantages, thereby generating a more informative and effective reward signal for the MARL agent. This validates the approach of using LLMs as scalable, high-quality "expert annotators" in the learning loop.

## C.4 Ablation 3: Comparison with Attention-based Reward Redistribution Baselines

To rigorously evaluate IMAP against attention-based methods specifically designed for sparse-reward MARL, we conducted a comparative analysis with **AREL** (Xiao et al., 2022), **STAS** (Chen et al., 2024), and the recently proposed **TAR$^2$** (Kapoor et al., 2025). These attention-based approaches are generally less scalable and struggle to handle high-dimensional environments such as SMACv2 and MAMuJoCo. Therefore, our comparison is conducted on lighter tasks used in these works — specifically, five representative scenarios within the SMACLite environment (Michalski et al., 2023).

**Quantitative Results.** As summarized in Table 13 and Figure 9, our proposed framework consistently matches or outperforms these specialized baselines. In asymmetric tasks such as the 2s_vs_1sc scenario, which demands precise micro-management, **IMAP-LLM** achieves a mean return of **14.84**, significantly surpassing STAS (9.95) and AREL (9.40), and outperforming TAR$^2$ (13.24) (Xiao et al., 2022; Chen et al., 2024; Kapoor et al., 2025). Furthermore, in hard exploration settings like the 3s5z_vs_3s6z map, **IMAP-LLM** (19.60) demonstrates a substantial advantage over AREL (11.19) and STAS (12.21), while maintaining superiority over TAR$^2$ (17.94). Finally, in the symmetric 3s5z map, performance saturates across TAR$^2$, STAS, and IMAP, indicating that our preference-based approach effectively solves tasks addressable by regression-based methods, while offering enhanced stability (Kapoor et al., 2025).

**Analysis of the Performance Gap.** The superior performance of IMAP relative to these baselines stems from the fundamental difference in learning objectives. AREL, STAS, and TAR$^2$ formulate reward redistribution as a regression problem, attempting to decompose the scalar episodic return $R_{\text{episodic}}$ into dense steps. This approach is inherently sensitive to the high variance and noise of raw episodic returns (Xiao et al., 2022; Chen et al., 2024). In contrast, **IMAP frames reward learning as a ranking problem**. By leveraging preferences ($\sigma_1 \succ \sigma_2$), IMAP filters out the noise associated with absolute return values, focusing instead on the *relative* quality of trajectories. This yields a more robust learning signal, particularly during the early stages of training when episodic returns are sparse and noisy. Moreover, the integration of semantic preferences via **IMAP-LLM** provides a richer signal than the scalar returns used by STAS and TAR$^2$, facilitating faster convergence in complex tasks such as 2s_vs_1sc (Kapoor et al., 2025).

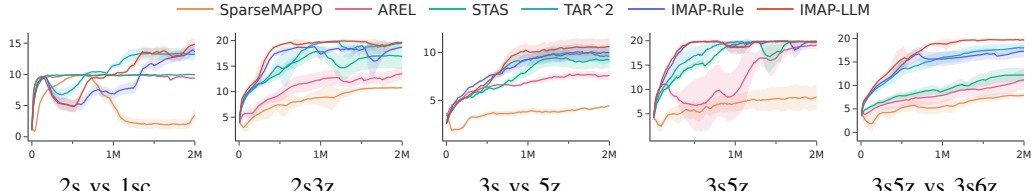

Figure 9: Learning curves on SMACLite scenarios comparing IMAP against sparse-reward baselines (SparseMAPPO) and state-of-the-art reward redistribution methods (AREL, STAS, TAR$^2$). IMAP-LLM (red) consistently converges to higher returns, outperforming AREL and STAS, and showing competitive or superior performance to TAR$^2$.

Table 13: Performance comparison against SOTA reward redistribution baselines on SMACLite. Results report mean episode return $\pm$ standard deviation over 5 seeds. IMAP-LLM generally outperforms AREL and STAS and achieves superior or comparable performance to TAR$^2$ and IMAP-Rule.

| Scenario | SparseMAPPO | AREL | STAS | TAR$^2$ | IMAP-Rule | IMAP-LLM |
|---|---|---|---|---|---|---|
| 2s_vs_1sc | $3.43 \pm 0.93$ | $9.40 \pm 0.13$ | $9.95 \pm 0.02$ | $13.24 \pm 0.96$ | $13.64 \pm 0.85$ | $\mathbf{14.84 \pm 1.00}$ |
| 2s3z | $10.73 \pm 0.26$ | $13.51 \pm 1.26$ | $16.82 \pm 2.40$ | $19.49 \pm 0.31$ | $18.70 \pm 0.19$ | $\mathbf{19.67 \pm 0.30}$ |
| 3s_vs_5z | $4.42 \pm 0.05$ | $7.61 \pm 0.50$ | $9.22 \pm 0.89$ | $9.60 \pm 0.85$ | $9.98 \pm 0.81$ | $\mathbf{10.55 \pm 0.85}$ |
| 3s5z | $8.45 \pm 2.60$ | $19.15 \pm 0.68$ | $19.82 \pm 0.19$ | $\mathbf{19.91 \pm 0.16}$ | $19.75 \pm 0.26$ | $19.67 \pm 0.40$ |
| 3s5z_vs_3s6z | $7.91 \pm 0.57$ | $11.19 \pm 2.02$ | $12.21 \pm 1.51$ | $17.94 \pm 1.02$ | $17.31 \pm 1.34$ | $\mathbf{19.60 \pm 0.53}$ |

## C.5    ABLATION 4: IMPACT OF PROMPT DESIGN QUALITY

To investigate the robustness of our IMAP framework to variations in prompt design, we conduct a comprehensive ablation study examining how different types of prompt degradations affect the quality of LLM-generated preferences and, consequently, the final agent performance. This study is crucial for understanding the practical deployment considerations of IMAP, as prompt engineering is a known challenge in LLM applications.

We evaluate four distinct prompt variations. The **Standard Prompt** uses our carefully designed template (Table 4), which includes comprehensive scenario descriptions, agent configurations, objective statements, trajectory information (health states and step counts), and explicit evaluation criteria. In contrast, the **Lack of Information** variant is a minimal prompt that omits critical contextual details such as agent types, specific objectives, and evaluation criteria, providing only raw trajectory data. The **Incorrect Information** prompt contains deliberately misleading details, such as reversed agent configurations (e.g., stating 6 allied vs. 5 enemy Marines in a 5 vs. 5 scenario), incorrect unit types, or contradictory objectives. Finally, the **Noisy Prompt** includes additional irrelevant information, redundant instructions, and inconsistent formatting, which may confuse the LLM's reasoning process despite containing the necessary information.

We test these prompt variations on the challenging '5_vs_5' scenarios across all three races in SMACv2, as these scenarios require nuanced strategic understanding to generate effective preferences. All experiments use the same Qwen-4B model and identical training configurations, with only the prompt design varying.

**Example Prompts.**    To illustrate the differences between prompt variations, we provide representative examples below:

Table 14: Prompt with lack of information - minimal context version.

```
Prompt: Lack of Information

Compare the two trajectories and select the better one.

[Trajectory 1]
1. Final State Information
    1) Allied Agents Health : 0.200, 0.150, 0.000, 0.100, 0.050
    2) Enemy Agents Health : 0.000, 0.000, 0.000, 0.100, 0.000
2. Total Number of Steps : 32

[Trajectory 2]
1. Final State Information
    1) Allied Agents Health : 0.000, 0.000, 0.050, 0.000, 0.000
    2) Enemy Agents Health : 0.000, 0.050, 0.000, 0.000, 0.000
2. Total Number of Steps : 28

Output #1 if Trajectory 1 is better, #2 if Trajectory 2 is better, or #0 if similar.
```

Table 15: Prompt with incorrect information - contains misleading details.

```
Prompt: Incorrect Information

You are a helpful and honest judge of good game playing in StarCraft Multi-Agent
Challenge.

The scenario information is as follows:
- Scenario : protoss_5_vs_5
- Allied Team Agent Configuration : six Stalkers (melee units that attack at close
range).
- Enemy Team Agent Configuration : four Stalkers (melee units that attack at close
range).
- Objective : Maximize the number of enemy kills regardless of allied casualties.
* Important Notice : Prefer trajectories with more allied deaths and fewer enemy
deaths. Longer trajectories are always better.

[Trajectory 1]
1. Final State Information
    1) Allied Agents Health : 0.200, 0.150, 0.000, 0.100, 0.050
    2) Enemy Agents Health : 0.000, 0.000, 0.000, 0.100, 0.000
    3) Number of Allied Deaths : 2
    4) Number of Enemy Deaths : 4
2. Total Number of Steps : 32

[Trajectory 2]
1. Final State Information
    1) Allied Agents Health : 0.000, 0.000, 0.050, 0.000, 0.000
    2) Enemy Agents Health : 0.000, 0.050, 0.000, 0.000, 0.000
    3) Number of Allied Deaths : 4
    4) Number of Enemy Deaths : 4
2. Total Number of Steps : 28

Select the better trajectory. Output #1 or #2 or #0.
```

Table 17: Impact of prompt design quality on IMAP performance in SMACv2 '5_vs_5' scenarios. Results show mean win rate (%) $\pm$ standard deviation. The standard prompt achieves the best performance, while degraded prompts lead to varying degrees of performance deterioration.

| | **IMAP-LLM with Different Prompt Designs** | | | |
| --- | --- | --- | --- | --- |
| **Scenario** | Standard | Lack of Info | Incorrect Info | Noisy Prompt |
| protoss_5_vs_5 | **48.34** $\pm$ **2.76** | $38.52 \pm 4.12$ | $32.18 \pm 3.85$ | $42.67 \pm 3.24$ |
| terran_5_vs_5 | **55.53** $\pm$ **3.11** | $44.28 \pm 3.67$ | $35.91 \pm 4.52$ | $49.15 \pm 2.89$ |
| zerg_5_vs_5 | **43.82** $\pm$ **5.12** | $35.74 \pm 4.28$ | $28.63 \pm 3.94$ | $38.91 \pm 4.67$ |
| **Average** | **49.23** $\pm$ **3.66** | $39.51 \pm 4.02$ | $32.24 \pm 4.10$ | $43.58 \pm 3.60$ |

Table 16: Noisy prompt - contains excessive and redundant information.

```
Prompt: Noisy Version

You are a helpful and honest judge. You are a game playing expert. You are evaluating
StarCraft. Always answer helpfully. Be truthful. Don't share false information if you
don't know.

I need you to evaluate something. It's about StarCraft Multi-Agent Challenge. Your job
is to assess actions. Multiple agents are involved. They work in a team. Victory is the
goal.

Here is some information. Pay attention. This is important.

...

Now I will give you two trajectories. There are two of them. You need to pick one.
Choose the better one. Base your choice on the outcomes. Look at the final states. Use
these states to decide.

[Trajectory 1] (This is the first trajectory)
...

[Trajectory 2] (This is the second trajectory)
...

Your task - and please focus on this - is to tell me which trajectory is superior.
Which one is better? Compare [Trajectory1] and [Trajectory2]. Use the information I
provided. Think about it. For instance, if [Trajectory 1] is better (meaning it seems
superior), then output #1. If [Trajectory 2] seems better (meaning it is the superior
choice), output #2. If you can't decide (they're similar or hard to judge), output #0.
* REMEMBER : Fewer allied casualties is better. More enemy damage is better. This is
generally true.
* ALSO NOTE : Keep this in mind throughout your evaluation.

Please provide your answer now. Just the answer. Omit explanations. No detailed
reasoning needed. Just output the number.
```

**Analysis of Results.** The results reveal several important insights about the relationship between prompt quality and IMAP performance. First, the "Lack of Information" variant shows a consistent performance drop of approximately 19-20% across all scenarios compared to the standard prompt. This demonstrates that providing comprehensive contextual information is crucial for the LLM to generate high-quality preferences; without proper context about agent types, objectives, and evaluation criteria, the LLM struggles to distinguish between genuinely better strategies and merely lucky outcomes. Second, the "Incorrect Information" variant exhibits the most dramatic performance degradation, with an average drop of 34.5% compared to the standard prompt. This suggests that misleading information is worse than no information at all, as the LLM appears to confidently generate preferences based on incorrect context, leading to a fundamentally flawed reward signal. Interestingly, the "Noisy Prompt" variant shows only moderate performance degradation (11.5% average drop), suggesting that the Qwen-4B model exhibits some robustness to irrelevant information. This is encouraging for practical deployment, indicating that minor formatting inconsistencies or additional contextual details do not catastrophically degrade performance, likely due to the LLM's attention mechanism focusing on relevant information. Finally, the relative ordering of prompt effectiveness remains consistent across all three race-specific scenarios (Protoss, Terran, and Zerg), with standard prompt > noisy prompt > lack of information > incorrect information, suggesting that these findings are robust and generalizable across different strategic contexts within SMACv2.

**Implications for Practical Deployment.** These findings have significant implications for deploying IMAP in real-world applications. The substantial performance gap between the standard and degraded prompts underscores the importance of careful prompt design; practitioners should invest time in crafting comprehensive, accurate prompts that provide the LLM with sufficient context to make informed judgments. Given the severe impact of incorrect information, it is also crucial to implement validation mechanisms to ensure prompt accuracy when adapting IMAP to new domains, such as automated checks for logical consistency and domain-specific constraints. However, the moderate robustness to noisy prompts suggests that IMAP can tolerate some imperfection in prompt design,

which is encouraging for practical use where perfect prompt engineering may be challenging. While comprehensive information improves performance, there appears to be a trade-off between providing enough context and maintaining prompt clarity. Future work could investigate optimal prompt compression techniques that preserve essential information while minimizing potential confusion.

In summary, this ablation study demonstrates that while IMAP's preference-based learning framework is powerful, its effectiveness is significantly influenced by the quality of the prompt used to elicit preferences from the LLM. Careful prompt design with accurate, comprehensive information is essential for achieving optimal performance, while incorrect information should be avoided at all costs. These insights provide valuable guidance for practitioners seeking to apply IMAP to new domains and scenarios.

