# OpenReview forum: "From Scarcity to Efficiency: Preference-Guided Learning for Sparse-Reward Multi-Agent Reinforcement Learning"
_ICLR.cc/2026/Conference — Submitted to ICLR 2026_

### Official Review · Reviewer_cDaS · 2025-10-27

**Soundness:** 2
**Presentation:** 1
**Contribution:** 2
**Rating:** 2
**Confidence:** 4

**Summary:**

This paper addresses the challenging problem of cooperative multi-agent reinforcement learning (MARL) in sparse-reward environments. The authors propose a novel framework, IMAP, that reframes the problem as one of online inverse preference learning (IPL). The core idea is to use the sparse, episodic reward (e.g., win/loss) not as a direct training signal, but as a source for generating trajectory-level preference labels (e.g., $\sigma_1 > \sigma_2$).

The framework then trains an implicit, decomposed reward model (based on a value-decomposition architecture) to learn dense, per-timestep rewards that are consistent with these preferences. This model then produces both a global reward signal ($R_t$) and agent-specific local reward signals ($r_{i,t}$). These dense rewards are then fed into a "dual-stream" policy gradient algorithm: the centralized critic is trained using a global advantage (from $R_t$), while the decentralized actors are trained using local advantages (from $r_{i,t}$) to improve credit assignment. The paper also explores using LLMs to generate more-nuanced preference labels. The authors provide theoretical analysis to argue that their learned reward function, while not the ground truth, converges to a behaviorally-equivalent function, thus preserving the optimal policy.

**Strengths:**

- **Significant and Relevant Problem**: The paper tackles a central and difficult challenge in MARL. Finding a way to perform effective credit assignment from sparse, episodic rewards is a highly significant research direction.

- **Novelty of the Core Idea**: The idea of integrating online inverse preference learning (IPL) directly with a value-decomposition network (VDN/QMIX-style) is novel. This is a creative combination of ideas from two different sub-fields (PbRL and MARL).

- **Promising Method for Credit Assignment**: The "dual-advantage" architecture derived from this decomposed reward model is an interesting and plausible approach to solving the multi-agent credit assignment problem. Using the learned local rewards for actors and the global reward for the critic is an elegant concept.

- **Exploration of LLM-based Feedback**: The (brief) exploration of using LLMs to provide more nuanced, qualitative preference feedback beyond simple win/loss signals is a timely and promising research avenue.

**Weaknesses:**

This paper, while built on promising ideas, suffers from major weaknesses in clarity, technical justification, and empirical evaluation. The work is not ready for publication and requires a major revision.

**Extremely Unclear Presentation**:

The paper is exceptionally difficult to follow. The writing is convoluted and fails to clearly explain the model's core mechanics.

- The "Dual-Critic" Architecture: The framework appears to use two separate sets of value functions: 1) the reward model's V-nets ($v_\xi$ and $M_w$), which we can call $V_{reward}$, and 2) the policy's separate centralized critic ($V_{\phi}^{tot}$), which we can call $V_{policy}$. Both are trained to estimate the exact same quantity: the value of a state under the current policy. This critical, redundant, and non-standard architectural choice is never explained or justified. The "Reward Learner" (Sec 4) already produces a complete, centralized value function ($V_{reward}$). A logical framework would reuse this as the policy critic. By introducing a second critic ($V_{policy}$) trained on the output of the first, the framework appears inefficient and suggests a poorly integrated design where a standard MAPPO implementation (which comes with $V_{policy}$) was simply "bolted onto" the reward-learning module (which produces $V_{reward}$).

- Convoluted Training Loop: The connection between the "Reward Learner" (Sec 4) and the "Policy Learner" (Sec 5) is poorly articulated. The alternating training process (training $q$ on preference loss, then $v$ on consistency loss, then $\pi$ on PPO loss) is a complex procedure that is not clearly detailed.

- Unclear Motivations: Key theoretical concepts are introduced without any motivation. For example, the Gumbel noise in Theorem 4.2 is stated as an assumption with no explanation for why it is the correct choice (i.e., that it is the theoretical justification for the Bradley-Terry model).

**Significant Technical and Theoretical Inconsistencies**:

- $\tau$ vs. $\beta$: The paper is sloppy with its notation. It uses $\beta$ as the temperature for the soft Bellman operator (Eq. 201) but introduces $\tau$ as the temperature for the Bradley-Terry model (Thm 4.2), setting $\tau=1$ by assumption. These parameters are conceptually identical. Both $\beta$ and $\tau$ function as the temperature parameter controlling the "softness" (or randomness) of an operation rooted in maximum entropy principles. $\beta$ controls the softness of the max in the LSE function ($V(s) = \beta \log \sum \exp(Q/\beta)$), while $\tau$ controls the softness of the choice in the BT model (which is a 2-item softmax: $P(A>B) = \exp(R_A/\tau) / (\exp(R_A/\tau) + \exp(R_B/\tau))$). A coherent theoretical framework would use a single, consistent temperature parameter. This discrepancy suggests a disjointed approach from different sources and a shallow theoretical understanding of the method.

- Theorem 4.2 Discrepancy: Theorem 4.2 is stated differently in the main text (line 252) and the supplementary material (line 785). This is a major red flag regarding the paper's carefulness and correctness.

- Confusing Proofs: As the authors' own pointers note, the proof of Theorem 4.1 (Appendix) contains a "confusing use of sigma" (line 815), making it difficult to verify.

- Proposition 4.1 is Not Novel: Proposition 4.1 (that an additive constant to trajectory-level reward does not change the optimal policy) is a long-established and well-known fact in reward-shaping literature (e.g., potential-based reward shaping). It is not a novel contribution of this paper.

**Unjustified Technical Assumptions**:

- The Shared Mixer Assumption: The paper's strongest assumption is that the global Q-function $Q_{tot}(s,a)$ and the global V-function $V_{tot}(s)$ share the **exact same mixing network** $\mathcal{M}_w$. This is a severe limiting assumption with no theoretical or empirical justification. Why should the mixing function for state-action values be identical to that of state values?

- LLM as a "Reasoner": The use of an LLM to "deduce preference rewards" (as noted in the pointers) is highly suspect. This assumes the LLM can reason about complex, non-trivial coordination dynamics in SMACv2 (or physics in MAMuJoCo, which was omitted). This is a very strong claim that is not substantiated, and the potential for the LLM to have pre-existing (training) knowledge of the environments is not addressed.

**Critically Insufficient Empirical Evaluation**:

This is the paper's most significant flaw. The authors claim to propose a new SOTA framework for sparse-reward MARL but fail to compare against any of the relevant, state-of-the-art baselines in this specific area.

- Missing Baselines: The experiments are missing comparisons to numerous critical works, including STAS [1], AREL [2], and TAR2 [3]. These methods are all directly designed for reward redistribution in sparse-reward MARL.

- Weak Baselines: The current baselines (SparseMAPPO, SL-MAPPO, Online-IPL) are not representative of the current state of the art.

- Without these comparisons, the empirical results in Table 1 are unsubstantiated and essentially meaningless. The paper does not demonstrate that its approach is any better than existing, simpler reward redistribution techniques.

[1] Chen, Sirui, et al. "STAS: spatial-temporal return decomposition for solving sparse rewards problems in multi-agent reinforcement learning." Proceedings of the AAAI Conference on Artificial Intelligence. Vol. 38. No. 16. 2024.

[2] Xiao, Baicen, Bhaskar Ramasubramanian, and Radha Poovendran. "Agent-temporal attention for reward redistribution in episodic multi-agent reinforcement learning." arXiv preprint arXiv:2201.04612 (2022).

[3] Kapoor, Aditya, et al. "$ TAR^ 2$: Temporal-Agent Reward Redistribution for Optimal Policy Preservation in Multi-Agent Reinforcement Learning." arXiv preprint arXiv:2502.04864 (2025).

**Questions:**

To facilitate a productive discussion, I ask the authors to please address the following points:

- Justify the Dual-Critic System: Please justify the redundant "dual-critic" architecture. Your reward-learning module (Sec 4) already produces a centralized value function, $V_{reward}$ = $M_w$ [$v_\xi$]. Why is this not used as the policy critic? Why is it necessary to introduce a second, separate centralized critic, $V_{\phi}^{tot}$, to be trained on the outputs of the first? This seems highly inefficient and implies a lack of integration between the two components of your framework.

- Justification for the Shared Mixer: Can you provide a clear theoretical or empirical justification for the assumption that $Q_{tot}(s,a)$ and $V_{tot}(s)$ can be factorized by the same mixing network? What is the impact of decoupling them?

- Missing Baselines: Can you justify the omission of critical and SOTA sparse-reward MARL baselines like STAS, AREL, and TAR2? A response to this is essential to evaluate the paper's empirical contribution.

- Temperature Inconsistency: Can you clarify the relationship between the temperature $\beta$ (soft Bellman) and the temperature $\tau$ (BT model)? Do you agree that they are conceptually identical, both controlling the "softness" of their respective MaxEnt operations? If so, why are they treated as separate, unconnected parameters?

- Theorem 4.2 Discrepancy: Can you please address and correct the discrepancy in the statement of Theorem 4.2 between the main text and the supplementary material?

---

> ### Author Response · Authors · 2025-11-26
>
> *We thank the reviewers for taking the time to read our paper and for providing insightful feedback. We have made our best effort to address all concerns and have updated the manuscript accordingly. Please find our detailed responses to each comment below, with the corresponding revisions highlighted in red within the paper.*
>
> > The "Dual-Critic" Architecture: The framework appears to use two separate sets of value functions:...
>
> We thank the reviewer for the insightful comment. To clarify, there are two sets of value functions produced in our framework — one in **Section 4** (preference learning) and another in **Section 5** (policy learning). The training of both these value functions is *not redundant*; it is in fact essential for our algorithm’s performance.
>
> In fact, the value functions obtained from preference learning are typically **inaccurate**, since preference learning does not interact with the environment and therefore fails to capture its underlying dynamics. Consequently, reusing these value functions for policy optimization would harm performance. This is consistent with our empirical findings: the **OnlineIPL** baseline, which relies on such preference-based value functions for policy extraction, performs substantially worse than our method.
>
> In contrast, our policy optimization method uses only the reward function derived from preference learning — which effectively redistributes the sparse reward signal — and incorporates it into the IMAP training process. This approach leverages the online learning mechanism of MAPPO to produce higher-quality policies. This combination of preference-learned rewards with MAPPO’s online policy learning is the key factor driving our strong performance.
>
> We have added a discussion in **Lines 351–359** of the revised manuscript to clarify this point. We hope this provides sufficient clarity, and we are happy to address any further questions or concerns if needed.
>
> > Convoluted Training Loop: The connection between the "Reward Learner" (Sec 4) and the "Policy Learner" (Sec 5) is poorly articulated ...
>
> We thank the reviewer for the insightful comment. As discussed above, our workflow proceeds as follows: we first perform preference learning to obtain an approximate dense reward function, where the weights $w_i$ represent each agent’s contribution to the global reward. This learned reward information is then used within our adapted MAPPO algorithm, which requires dense rewards for effective policy optimization.We have added a discussion in **Section 5.2** to clarify that only the learned reward information—rather than the full value functions—is transferred from the preference-learning stage to the policy-learning stage (MAPPO with dual advantages). **Algorithm 1** in the Appendix further illustrates this process. We would be happy to provide additional clarification if the reviewer could kindly indicate which specific aspects remain unclear.
>
> > Unclear Motivations: Key theoretical concepts are introduced without any motivation. For example, the Gumbel noise in Theorem 4.2 ...
>
> We thank the reviewer for the insightful question. To clarify, in Theorem~4.2, we introduce Gumbel-distributed noise to the aggregated trajectory rewards, which makes the preference between two trajectories $\sigma_1$ and $\sigma_2$ stochastic and follows the Bradley–Terry model. This probabilistic formulation ensures that the induced preference distribution aligns with the BT likelihood, which is essential for establishing the \textit{asymptotic convergence} result. Importantly, this stochastic assumption serves a purely theoretical role—our practical implementation does not explicitly inject noise but naturally approximates this behavior through inherent environmental and sampling randomness.
>
> *We have added a clarification below the theorem to make this motivation explicit.*
>
> > $\tau$ and $\beta$, these parameters are conceptually identical ...
>
> We thank the reviewer for the insightful question. In our formulation, we believe that that the two parameters are **not conceptually identical**, i.e.,  $\tau$ in the BT model and  $\beta$ in the soft Bellman equation should serve distinct theoretical purposes and should not be treated as equivalent. Specifically, $\tau$ governs the stochasticity of the preference model, controlling how sensitively the model differentiates between trajectory rewards, while $\beta$ regulates the entropy regularization strength and thus the smoothness of the resulting soft policy. These parameters influence different components of the learning process—preference modeling versus policy optimization—and are not mathematically interchangeable. Therefore, we believe it would be inappropriate to treat them as identical in our formulation.
>
> We hope our explanation clarifies why it is necessary to maintain separate parameters for the BT model and the soft Bellman equation. We remain open to further discussion on this point should the reviewer deem it necessary.

---

> > ### Author Response · Authors · 2025-11-26
> >
> > > Theorem 4.2 Discrepancy: Theorem 4.2 is stated differently in the main text (line 252) and the supplementary material (line 785)
> >
> > We thank the reviewer for the observation. The version of the theorem in the Appendix is indeed the same as that in the main paper, but presented with slightly more detail. Initially, the full version appeared in the main text; however, we shortened it to comply with the 9-page limit. We have now updated the **Appendix** to ensure that the theorem statement is fully consistent with the one presented in the main paper.
> >
> > > Confusing Proofs: As the authors' own pointers note, the proof of Theorem 4.1 (Appendix) contains a "confusing use of sigma" (line 815), making it difficult to verify.
> >
> > Thank you for the observation. To address this, we have updated the notation of the logistic function to **\Delta** to clearly distinguish it from the trajectory notation.
> >
> > > Proposition 4.1 is Not Novel:..
> >
> > Thank you for the insightful comment, with which we agree. We have now removed the proposition and instead cited a relevant paper that formally states the reward-shaping result.
> >
> > > The Shared Mixer Assumption: The paper's strongest assumption is that the global Q-function  and the global V-function  share the exact same mixing network . ...
> >
> > We thank the reviewer for the insightful question. To clarify, in our mixing architecture, $w_i$ represents the contribution weight of each agent $i$ to the global value functions. Thus, for consistency, we employ **the same set of parameters $\{w_i\}$** for both $Q_{\mathrm{tot}}$ and $V_{\mathrm{tot}}$. This design choice is also commonly adopted in other MARL works utilizing value decomposition architectures.
> >
> > We have added a clarification in **Lines 217–221** of the revised manuscript to make this point explicit.
> >
> > > LLM as a "Reasoner": The use of an LLM to "deduce preference rewards" (as noted in the pointers) is highly suspect.
> >
> > We thank the reviewer for the thoughtful comment. We agree that relying on an LLM to fully reason about complex coordination dynamics would be an overly strong claim, and we wish to clarify that this is *not a major component* of our work. The LLM-based component is **not a core contribution**, but rather a demonstration of the flexibility of our framework to incorporate semantic or human-aligned preference information when interpretable features are available (e.g., in SMACv2). The LLM serves only as an auxiliary preference labeler, providing coarse comparative judgments between trajectories rather than detailed reasoning about environment dynamics.
> >
> > We also note that IMAP performs effectively without the LLM component, as shown by the strong results of IMAP-Rule on both SMACv2 and MAMuJoCo. This indicates that our core contribution lies in the implicit reward learning and dual-advantage optimization framework, rather than the LLM itself.
> >
> > Regarding the possibility of the LLM having pre-existing knowledge of the environments, we note that our use of the LLM does not rely on or exploit any environment-specific prior information. The model only receives structured textual summaries generated from our simulation trajectories (e.g., unit statistics or outcomes) without any explicit contextual details about SMACv2 or its map configurations. Moreover, we use a locally run Qwen2-4B model without Internet access, ensuring that no external or environment-specific retrieval occurs during training.
> >
> > In line with your point, we can note that, for other environments such as MAMuJoCo, or domains lacking interpretable features, a pre-trained or fine-tuned LLM specialized for that environment might be required—an interesting direction we leave for future research.
> >
> > > The authors claim to propose a new SOTA framework for sparse-reward MARL but fail to compare against any of the relevant, state-of-the-art baselines in this specific area
> >
> > We thank the reviewer for suggesting these baselines, which we find highly relevant. To address this point, *we have included additional comparisons* with the suggested attention-based methods in **Appendix C.4** of the revised manuscript. We note that these methods are generally not scalable to high-dimensional environments such as SMACv2 and MAMuJoCo considered in our paper. Therefore, we follow their original experimental settings and conduct comparisons on SMACLite, a lightweight version of SMAC used in those works. We believe this addition provides a fair and meaningful comparison, further strengthening the empirical evaluation of our approach.
> >
> > **We sincerely appreciate the reviewer’s detailed and constructive feedback. We hope that our responses and revisions are convincing, particularly with the inclusion of additional attention-based baselines. We thank you again for your valuable comments, which have significantly improved the quality of our paper. Should you have any further questions or suggestions, we would be happy to provide additional clarification or discussion.**

---

### Official Review · Reviewer_aXhn · 2025-10-29

**Soundness:** 3
**Presentation:** 3
**Contribution:** 3
**Rating:** 6
**Confidence:** 4

**Summary:**

The paper tackles sparse-reward cooperative MARL by unifying inverse preference learning (IPL) with a CTDE, PPO-style learner. It 1. converts episodic returns into trajectory preferences, 2. learns an implicit reward via a preference-based value-decomposition network and 3. trains with dual advantage streams—a global advantage for the centralized critic and agent-specific local advantages for decentralized actors. It also explores using LLMs to label preferences from trajectory summaries. Experiments on SMACv2 and MAMuJoCo report stronger win-rates/returns and better sample efficiency than baselines

**Strengths:**

The idea is quite interesting and simple. The decomposition gives local advantages aligned with the global objective, addressing a common weakness of MAPPO with a single global signal.

The theoretical analysis shows that the learned implicit reward converges to a behaviorally indistinguishable surrogate of the ground-truth

The formulation is quite clear and the writing is easy to follow.

**Weaknesses:**

The theoretical part seems to strong: Theorem 4.2 assumes BT-generated preferences with τ=1, coverage over all trajectory pairs, and N ⁣→ ⁣∞. In practice we only have finite samples. How do you ensure adequate pairwise coverage under realistic budgets, and what is the sampling cost (queries, wall-clock, tokens) required to achieve the stated guarantees? Also, as we may not be dealing with fully converged cases, how sensitive is performance to semi-coverage?

While effective, LLM labels add potential bias and online cost; Can you quantify the LLM-preference budget (tokens/time) and report win-rate per dollar or per wall-clock hour to substantiate efficiency claims? Also, you seem to be missing the IMAP-LLM for MA-MOJOCO. Any reason for not including the result for that one?

**Questions:**

See weakness.

---

> ### Author Response · Authors · 2025-11-26
>
> *We thank the reviewers for taking the time to read our paper and for providing insightful feedback. We have made our best effort to address all concerns and have updated the manuscript accordingly. Please find our detailed responses to each comment below, with the corresponding revisions highlighted in red within the paper.*
>
> > The theoretical part seems to strong: Theorem 4.2 assumes BT-generated preferences with τ=1, coverage over all trajectory pairs, and N ⁣→ ⁣∞. In practice we only have finite samples. How do you ensure adequate pairwise coverage under realistic budgets, and what is the sampling cost (queries, wall-clock, tokens) required to achieve the stated guarantees? Also, as we may not be dealing with fully converged cases, how sensitive is performance to semi-coverage?
>
> We thank the reviewer for this thoughtful question. Here we would like to clarify that Theorem~4.2 presents an *asymptotic* result, meaning that convergence is guaranteed only as the number of sampled trajectory pairs approaches infinity. Similar to other asymptotic analyses in preference-based or inverse RL, this theoretical assumption does not strictly hold in practice but provides insight into how the learned reward function approaches the true underlying reward as the data scale increases. In realistic settings, achieving full or even partial coverage of all trajectory pairs is infeasible; however, it is not necessary to recover the exact true reward. Learning an approximate reward distribution that preserves relative preferences among trajectories is sufficient to induce high-quality policies. This is empirically supported by our experiments, where IMAP converges to strong-performing policies with a relatively small number of collected preference pairs.
>
> > While effective, LLM labels add potential bias and online cost;
>
> We thank the reviewer for the comment, with which we agree. To clarify the potential impact of bias introduced by the LLM, we have already included an ablation study comparing different LLM models (**Appendix C3**). In addition, we have added new results in **Appendix C.5** analyzing the effect of prompt design on the performance of our algorithm.
>
> > Can you quantify the LLM-preference budget (tokens/time) and report win-rate per dollar or per wall-clock hour to substantiate efficiency claims?
>
> We thank the reviewer for this question. In our implementation, the LLM (Qwen2-4B) runs locally and is lightweight, so the additional computational overhead is minimal. As such, reporting token or monetary costs—typically relevant for API-based models—is not particularly informative in our setting. The LLM labeling constitutes only a small fraction of the total training time, and its contribution remains consistent across runs.We have added a note discussing the computational cost of LLM calls in our experiments (lines 449–501 of the revised manuscript). We hope this clarification is sufficient, and we would be happy to provide additional details if the reviewer finds it necessary.
>
> > Also, you seem to be missing the IMAP-LLM for MA-MOJOCO. Any reason for not including the result for that one?
>
> We thank the reviewer for the question. The IMAP-LLM variant is not included for the MAMuJoCo experiments because the LLM-guided preference module relies on interpretable, high-level features (e.g., cooperative behaviors, spatial formations, or task-level outcomes) that are available in discrete multi-agent settings such as SMACv2, but not in continuous-control domains like MAMuJoCo. In MAMuJoCo, the state and action representations are high-dimensional and lack semantically meaningful descriptions that an LLM can reliably interpret. For this reason, we evaluate IMAP-Rule in MAMuJoCo as a more appropriate and fair variant for continuous control. We have clarified this rationale in **Section 6** of the revised manuscript.
>
> **We hope that our responses and the corresponding clarifications to the manuscript adequately address the reviewer’s concerns. We sincerely appreciate the constructive feedback and are happy to provide further clarification or additional analysis if needed.**

---

### Official Review · Reviewer_e5ym · 2025-11-01

**Soundness:** 3
**Presentation:** 3
**Contribution:** 3
**Rating:** 6
**Confidence:** 4

**Summary:**

This paper primarily addresses the sparse reward and credit allocation problem in MARL with episodic terminal rewards. The core idea is to construct trajectory preferences based on the relative ranking of trajectory rewards, and then recover Q-values ​​from the trajectories using the inverse soft Bellman operator of IPL for value decomposition. Building upon MAPPO, dual advantage streams are introduced to achieve finer credit allocation, with corresponding policy enhancement theoretical guarantees provided. Furthermore, for scenarios with clearly structured preference information, LLM is used to construct denser preference signals, further improving the scalability of this method in complex tasks without explicit rewards. Experiments on multiple MARL benchmark tasks effectively demonstrate the performance of the proposed method.

**Strengths:**

This paper's motivation for applying preference learning to online sparse reward MARL problems is sound. Building upon IPL, it combines value decomposition methods to effectively achieve credit allocation among agents.

Using the inverse soft Bellman operator to recover rewards in Q-space effectively avoids the neglect of environmental state transitions in simple BT models, making it more suitable for value-based RL methods. The consistency of global-local advantages provides a theoretical guarantee for this approach.

This paper considers discrete and continuous control task environments, conducting reasonable baseline comparisons and ablation experiments. It also provides a comparison between rule-based preferences and LLM preferences, demonstrating the potential of LLM preferences in real-world scenarios where dense rewards are difficult to design.

**Weaknesses:**

W1: The paper defines $r_i (s_i, a_i) = q_i(s_i, a_i) − \gamma E_{p(s’_i|s_i,a_i)}[v_i(s’_i)]$  for each agent. However, in partially observable, coupled dynamical cooperative environments, local state transitions $p(s’_i | s_i, a_i)$ generally depend on the actions of other agents and the global state, making the "local inverse Bellman" assumption somewhat too strong. Especially for the POMDP problem using local observations as input, does this property still hold? It is suggested that the authors further explain what approximations were used in the implementation to handle the local observation problem.

W2: In terms of experiments, the comparison with policy gradient methods with centralized critics like MAPPO is insufficient to fully demonstrate the advantages in credit assignment with sparse rewards, such as FACMAC and HAPPO.

W3: MAMuJoCo tasks are generally dense rewards by default. The article claims a "sparse reward setting" but does not clearly explain how to modify it to only trajectory-level feedback. More specific environmental settings are needed.

**Questions:**

Q1: A fundamental question with PBRL is how to handle the learning process when early stochastic policies offer similar trajectory rewards or lack exploration, resulting in very little preference information. Are cold-start techniques employed?

Q2: How to extract the corresponding $wi*$ from the value decomposition network of a multi-layer MLP for calculating $\delta_{i,t}^{local}$?

Q3: What are the specific decoding parameter settings for LLM? For example, temperature and top p, and is thinking mode enabled on qwen3-4b?

Q4: This paper only considers constructing the LLM preference reward using structured text to describe the final state, which is merely the final state preference. A natural idea is to input the complete trajectory policy into the LLM to evaluate the trajectory preference. Can the method presented in this paper be extended to this setting?

---

> ### Author Response · Authors · 2025-11-26
>
> *We thank the reviewers for taking the time to read our paper and for providing insightful feedback. We have made our best effort to address all concerns and have updated the manuscript accordingly. Please find our detailed responses to each comment below, with the corresponding revisions highlighted in red within the paper.*
>
> > W1: The paper defines $r_i(s_i,a_i)$ for each agent. However, in partially observable, coupled dynamical cooperative environments, local state transitions generally depend on the actions of other agents and the global state...
>
> We thank the reviewer for the insightful comment. We agree that the local transition probability should depend on the joint state and joint action, rather than only the local ones. Accordingly, we have updated the definition of the local reward as a function of the local $q$ and $v$ values to reflect this dependency.
>
> In practice, we only have access to local observations (as noted earlier, the use of $\mathbf{s}$ and $s_i$ in the main text is for notational simplicity). We do not sample local actions or observations independently; instead, we sample joint actions and extract the corresponding local components to estimate the local value functions. The full details of this implementation are provided in the algorithm in Appendix B3, where we describe how global observations and joint actions are sampled and stored, and how the local value functions are updated. We hope our response provides sufficient clarity, and we are happy to include a more detailed explanation if desired.
>
> > W2: In terms of experiments, the comparison with policy gradient methods with centralized critics like MAPPO is insufficient to fully demonstrate the advantages in credit assignment with sparse rewards, such as FACMAC and HAPPO.
>
> We thank the reviewer for the valuable comment. To address, we have now included additional comparisons with **HAPPO**, a strong and more recent policy-gradient method with a centralized critic, in our updated experiments. The results (see Tables 1–2) show that IMAP consistently outperforms HAPPO across all tested scenarios, demonstrating its effectiveness in credit assignment under sparse-reward conditions. We acknowledge that FACMAC is another relevant method; however, it is relatively older and primarily designed for dense-reward control tasks, making it less comparable to our sparse-reward setting.
>
> > W3: MAMuJoCo tasks are generally dense rewards by default. The article claims a "sparse reward setting" but does not clearly explain how to modify it to only trajectory-level feedback. More specific environmental settings are needed.
>
> We thank the reviewer for the insightful comment. While MAMuJoCo is originally a dense-reward environment, we modify it to align with our sparse-reward setting. Specifically, instead of using per-step rewards, agents receive a single *trajectory-level* reward at the end of each episode, computed as the normalized cumulative return over the full trajectory. To clarify this design choice, we have added detailed explanations in **Appendix B.1.2** of the revised manuscript. We hope this clarification addresses the reviewer’s concern.
>
> > Q1: A fundamental question with PBRL is how to handle the learning process when early stochastic policies offer similar trajectory rewards or lack exploration, resulting in very little preference information. Are cold-start techniques employed?
>
> We thank the reviewer for this important question. Indeed, in the early training phase, stochastic policies may produce trajectories with similar cumulative rewards, resulting in weak or uninformative preference signals. To mitigate this cold-start issue, IMAP employs two complementary strategies. First, we initialize the preference dataset using a small set of randomly sampled trajectories to ensure basic reward diversity. Second, we leverage the rule-based preference module (IMAP-Rule) in the initial stages to provide coarse but informative guidance before switching to learned or LLM-assisted preference models as policies improve. This hybrid initialization helps bootstrap learning and encourages exploration, ensuring that preference information becomes increasingly discriminative as training progresses. We have added a clarification of this mechanism in **Appendix B3.1** of the revised manuscript.
>
> > Q2: How to extract the corresponding  from the value decomposition network of a multi-layer MLP for calculating?
>
> We thank the reviewer for the question. In our setting, the mixing network is linear, and $w_i$ represents the learned weights of this linear mixing function. In practice, these weights are modeled as outputs of a small hypernetwork conditioned on local information. After the preference-based learning stage, the values of $w_i$ can be directly extracted from the trained network parameters.
>
> (As a side note, the MLP is only employed only in the actor and critic networks during the policy optimization stage.)

---

> > ### Author Response · Authors · 2025-11-26
> >
> > > Q3: What are the specific decoding parameter settings for LLM? For example, temperature and top p, and is thinking mode enabled on qwen3-4b?
> >
> > We thank the reviewer for the question. The LLM component (Qwen2-4B-Instruct) was used in standard inference mode. We did not perform prompt-level hyperparameter tuning; instead, we used the model’s default decoding configuration as provided by the official release, without enabling the “thinking” mode. We have added this clarification to **Appendix B.3** of the revised manuscript.
> >
> > > Q4: This paper only considers constructing the LLM preference reward using structured text to describe the final state, which is merely the final state preference. A natural idea is to input the complete trajectory policy into the LLM to evaluate the trajectory preference. Can the method presented in this paper be extended to this setting?
> >
> > We thank the reviewer for the insightful question. We agree that extending the LLM-based module to evaluate full trajectories is a natural direction. In this work, following recent studies on LLM-supported MARL (Bui and et 2025), we focus on final-state descriptions for better clarity, interpretability, and computational efficiency. Evaluating entire trajectories would require processing long sequential inputs, which may introduce noise and significant computational overhead, which is challenging in our online learning. Nevertheless, we note that our framework is fully compatible with trajectory-level preference modeling if computational resources permit, and we consider this a promising direction for future work.
> >
> > **We hope that our responses and the corresponding updates to the manuscript adequately address the reviewer’s concerns. We sincerely appreciate the constructive feedback and are happy to provide further clarification or additional analysis if needed.**

---

### Official Review · Reviewer_C2ir · 2025-11-10

**Soundness:** 2
**Presentation:** 3
**Contribution:** 2
**Rating:** 4
**Confidence:** 5

**Summary:**

This paper addresses the challenge of cooperative multi-agent reinforcement learning in environments with sparse rewards, where feedback is only provided at trajectory termination. The authors propose IMAP, which converts sparse episodic rewards into trajectory preferences and learns implicit dense rewards through inverse preference learning in the Q-function space. Key contributions include transforming sparse rewards into preferences, a CTDE paradigm with dual advantages, theoretical guarantees on reward convergence and policy alignment, and evaluations on MAMuJoCo and SMACv2 benchmarks showing superior performance and efficiency.

**Strengths:**

- The paper tackles the highly challenging and relevant problem of sparse rewards in online cooperative MARL. This work successfully connects recent advances in offline preference-based RL with the online, multi-agent CTDE paradigm, resulting in a unified and novel learning framework.
- The introduction of dual advantage streams is a key contribution for credit assignment in the sparse-reward setting. By using a global advantage for the centralized critic and distinct local advantages for the decentralized actors, the method effectively separates global coordination from local credit assignment.
- Proposition 4.1 and Theorem 4.2 establish the asymptotic convergence of the learned implicit reward. The key result is that the method converges to a reward function that is "behaviorally indistinguishable" from the ground-truth, ensuring the optimal policy aligns with the true task objective.
- The empirical results are very strong. IMAP-Rule and IMAP-LLM significantly outperform all baselines.

**Weaknesses:**

- The entire value decomposition framework relies on a simple linear mixing network $\mathcal{M}_w$. This is a strong assumption and is significantly less expressive than the non-linear, monotonic mixing networks used in ubiquitous MARL algorithms like QMIX. The paper provides no justification for this choice or analysis of whether this restrictive structure limits performance.
- The LLM-based component relies on "interpretable features". This is why it is applied to SMACv2 but not MAMuJoCo. This restriction significantly limits the generality of the paper's most novel-looking variant, as many complex problems (like continuous control) do not have easily interpretable, human-readable features.
- The success of IMAP-LLM shifts the burden of "reward engineering" to "prompt engineering". The performance is likely sensitive to the quality of the prompt, such as the "Important Notice" and "Objective". The paper lacks a sensitivity analysis on the prompt's content, making it unclear how much expert knowledge is implicitly required to craft an effective prompt.
- The definition of the local implicit reward $r_i$ in Sec. 5.2 is confusing. The local transition $P(\cdot|s_{i},a_{i})$ is generally not well-defined in a multi-agent setting, as the next state $s'$ depends on the joint action, not just $a_i$. This notation seems to conflict with the CTDE assumptions.
- Theorem 4.2 assumes every possible trajectory pair appears in the preference dataset at least N times, which is impractical for high-dimensional state-action spaces or long horizons.
- No comparison is provided against recent sparse-reward baselines specifically designed for MARL, such as attention-based return decomposition methods mentioned in Section 2 but absent from experiments.
- Notation for global state s sometimes interchanges with joint observation o without consistent clarification.

**Questions:**

- Theorem 4.2 assumes every possible trajectory pair appears at least N times in the preference dataset. How does your online algorithm (Algorithm 1) ensure sufficient coverage in practice?
- How do you manage the preference buffer P in Algorithm 1? Do preferences expire, and how is sampling performed?
- What is the actual computational overhead of using LLMs for online preference generation, and when does the performance gain justify this cost?
- How sensitive is IMAP to key hyperparameters, particularly the temperature $\beta$, regularization $\varphi$, and the number of IPL gradient steps per iteration?
- How sensitive is IMAP-LLM to prompt design? Have you tested alternative prompts or prompt templates?
- The method generates preferences by sampling and comparing pairs of trajectories. Could you comment on the sample efficiency of this "tournament-style" sampling? Does this pairwise comparison loop introduce a significant sample complexity overhead compared to methods that can learn from individual trajectory returns?

---

> ### Author Response · Authors · 2025-11-26
>
> *We thank the reviewers for taking the time to read our paper and for providing insightful feedback. We have made our best effort to address all concerns and have updated the manuscript accordingly. Please find our detailed responses to each comment below, with the corresponding revisions highlighted in red within the paper.*
>
> > The entire value decomposition framework relies on a simple linear mixing network ...
>
> We thank the reviewer for the valuable comment! We would like to note that our use of a linear mixing network is a deliberate design choice rather than a simplification. Recent MARL studies on complex multi-agent tasks including SMACv2 and MuJoCo have shown that non-linear mixing architectures often lead to overfitting and lower sample efficiency compared to simpler linear structures (Bui et all. (2025)). Moreover, the linear formulation preserves an *additive* decomposition between global and local values, which is crucial for constructing agent-specific advantages that remain theoretically consistent with the global objective (our Propositions~1–2).
> In contrast, non-linear monotonic mixers such as QMIX, while more expressive, obscure this additivity and hinder the extraction of interpretable local rewards required for our dual-advantage PPO framework.
>
> To clarify, we have added a discussion justifying this design choice in lines~[216–225] of the revised manuscript.
>
> > The LLM-based component relies on "interpretable features". This is why it is applied to SMACv2 but not MAMuJoCo. This restriction significantly limits the generality of the paper's most novel-looking variant,  ...
>
> We thank the reviewer for the insightful comment, with which we agree. The LLM-based component indeed relies on interpretable trajectory features, making it more suitable for environments such as SMACv2 where high-level behavioral descriptors (e.g., unit formation, health, and coordination) are available. However, we emphasize that the LLM-guided preference module is *not the core contribution of our work*, but rather a demonstration of the *flexibility* of our preference-based framework to incorporate semantic or human-aligned supervision when such features exist. The main contribution of IMAP lies in its unified implicit reward learning and dual-advantage optimization framework, which performs effectively even without LLM-based guidance.
>
> > `How sensitive is IMAP-LLM to prompt design?
>
> We appreciate the reviewer’s thoughtful question. We agree that the LLM-guided component may introduce sensitivity to prompt design. To address this, we have conducted an *additional sensitivity analysis* on prompt content under various scenarios, including missing, incorrect, and noisy information. The results, now presented in Appendix~C.5 of the revised manuscript, demonstrate that IMAP-LLM remains robust across these conditions, showing only minor performance fluctuations.
>
> > The definition of the local implicit reward  in Sec. 5.2 is confusing...
>
> We thank the reviewer for the thoughtful observation. We have updated the definition of the local reward in Section~5.2 to explicitly reflect its dependency on the joint state and joint actions.
>
> > Theorem 4.2 assumes every possible trajectory pair appears in the preference dataset at least N times
>
> We appreciate the reviewer’s observation. We agree that the assumption in Theorem~4.2 is indeed asymptotic and not directly achievable in high-dimensional or long-horizon settings. While this does not hold strictly in practice, it provides valuable theoretical insight into the convergence behavior of our framework under idealized coverage. In realistic settings, our online iterative learning process (with a finite preference buffer) serves as a practical approximation, allowing the model to adapt continually to the evolving trajectory distribution.
>
>  > No comparison is provided against recent sparse-reward baselines specifically designed for MARL, such as attention-based..
>
> We thank the reviewer for the insightful comment. To address this, we have added additional comparisons with attention-based methods in Appendix~C.4. We note that such approaches are generally not scalable to high-dimensional environments like SMACv2 or MAMuJoCo. Therefore, following prior works, we conduct these comparisons using the lightweight SMACLite environment.
>
> > Notation for global state s sometimes interchanges with joint observation o without consistent clarification
>
> We thank the reviewer for the comment. As noted in lines 143–144, we use $\mathbf{s}$ in the main text for notational simplicity, while acknowledging that, in practice, each agent only has access to its local observation, denoted by $\mathbf{o}$. Accordingly, when describing the practical implementation details in Appendix B.2, we switch to the $\mathbf{o}$ notation to reflect this setting. We have also added a clarifying sentence in Appendix B.2 of the revised manuscript.

---

> > ### Author Response · Authors · 2025-11-26
> >
> > > How do you manage the preference buffer P in Algorithm 1? Do preferences expire, and how is sampling performed?
> >
> > We thank the reviewer for the question. We maintain a FIFO buffer to store and sample preference data during training. To clarify this implementation detail, we have added explanatory sentences in lines [277–280] of the revised manuscript. We can further expand this description if additional detail is desired.
> >
> > > What is the actual computational overhead of using LLMs for online preference generation, and when does the performance gain justify this cost?
> >
> > We thank the reviewer for the question. We use Qwen-4B for generating preferences, which is a lightweight model capable of running locally and is well-suited for our online training setting. We have added clarifications regarding the computational overhead in lines [496–498] of the revised manuscript. We hope this provides sufficient clarity, and we are happy to include further implementation details if desired.
> >
> > > How sensitive is IMAP to key hyperparameters, particularly the temperature , regularization , and the number of IPL gradient steps per iteration?
> >
> > We thank the reviewer for this important question. We base our default settings on well-established choices from prior works in multi-agent IPL and MAPPO, where these parameters have been carefully analyzed. In practice, we found that moderate changes in temperature, regularization strength, and the number of IPL gradient steps per iteration lead to consistent performance, indicating that IMAP does not require extensive hyperparameter tuning.
> >
> > > The method generates preferences by sampling and comparing pairs of trajectories. Could you comment on the sample efficiency of this "tournament-style" sampling? Does this pairwise comparison loop introduce a significant sample complexity overhead compared to methods that can learn from individual trajectory returns?
> >
> > We thank the reviewer for the insightful question. While IMAP employs pairwise trajectory comparisons, the additional sampling cost is minimal because preferences are derived from the same trajectories already collected for policy updates—no extra environment interactions are required. The “tournament-style” comparison thus reuses existing experience data, introducing only a negligible computational overhead. Moreover, preference-based supervision often provides denser and more informative learning signals than sparse scalar returns, improving sample efficiency overall. Empirically, as shown in Tables 1–3, IMAP achieves faster convergence and higher final performance than return-based baselines such as SparseMAPPO and SL-MAPPO, demonstrating that the preference-learning loop enhances, rather than hinders, sample efficiency.
> >
> > **We hope that our responses and the corresponding updates to the manuscript adequately address the reviewer’s concerns. We sincerely appreciate the constructive feedback and are happy to provide further clarification or additional analysis if needed.**

---

### Author Response · Authors · 2025-11-28
**Summary of authors's responses**

***Dear AC and Reviewers, to give an overview of our response and updates we have made to the paper, we summarize the major points of our responses below.***

- **Reviewer C2ir** comments about **the use of a simple linear mixing network**. To address we clarify that the linear network is a deliberate design choice that preserves the **additive decomposition** crucial for agent-specific advantages and is theoretically consistent with the global objective, unlike non-linear mixers which obscure this property. We added a clarification to paper at lines **[216–225]**

- **Reviewer C2ir** comments about **the lack of comparison against recent sparse-reward baselines** like attention-based return decomposition methods. To address we have added additional comparisons with these attention-based methods in the **SMACLite** environment. We made updates to paper at **Appendix C.4**.

- **Reviewer C2ir** comments about **the sensitivity of IMAP-LLM to prompt design**. To address we conducted an additional sensitivity analysis on prompt content under various scenarios (missing, incorrect, and noisy information), reported in  **Appendix C.5**

- **Reviewer C2ir** concerns about **the definition of the local implicit reward in Sec. 5.2**. To address we have updated the definition of the local reward in Section 5.2 to explicitly reflect its dependency on the **joint state and joint actions**.

- **Reviewer e5ym** comments about **insufficient comparison with policy gradient methods with centralized critics** like HAPPO. To address we have included additional comparisons with **HAPPO** in our updated experiments. We made updates to paper at **Tables 1–2**.

- **Reviewer e5ym** comments about the **MAMuJoCo sparse reward setting** not being clearly explained. To address we added a clarification to paper at **Appendix B.1.2** of the revised manuscript.

- **Reviewer e5ym** comments about the **cold-start issue in PBRL** where early stochastic policies offer little preference information. To address we clarify that we employ two complementary strategies: initializing the preference dataset with a small set of randomly sampled trajectories, and leveraging the **rule-based preference module (IMAP-Rule)** in the initial stages before switching to learned models. We made updates to paper at **Appendix B 3.1**

- **Reviewer e5ym** comments about the **specific decoding parameter settings for the LLM**. To address we clarify that the LLM component (Qwen2-4B-Instruct) was used in **standard inference mode** with the model’s **default decoding configuration** and without enabling the "thinking" mode. We made updates to paper at **Appendix B.3**.

- **Reviewer aXhn** comments about **missing IMAP-LLM results for MA-MuJOCO**. To address we clarify that IMAP-LLM was omitted for MAMuJoCo because the LLM-guided preference module relies on **interpretable, high-level features** available in SMACv2 but not in continuous-control domains like MAMuJoCo. We added a clarification to paper at **Section 6**

- **Reviewer cDaS** comments about the **redundant "Dual-Critic" Architecture**. To address we clarify that the two sets of value functions are **not redundant**; the preference-learned value functions are often **inaccurate** as they do not interact with the environment. Therefore, we use a separate centralized critic (MAPPO's) for policy optimization, only transferring the **learned reward function** (not the value functions) from the preference-learning stage. We made updates to paper at lines **351–359** of the revised manuscript and clarified the transfer mechanism in **Section 5.2**.

- **Reviewer cDaS** comments about the **Unclear Motivations for Gumbel noise in Theorem 4.2**. To address, we clarify that the Gumbel-distributed noise is introduced for purely **theoretical** purposes to make the preference stochastic and follow the **Bradley–Terry model**. We made updates to paper by adding a clarification **below the theorem**.

*We believe that we have thoroughly addressed all major concerns raised by the reviewers and have made substantial updates to the manuscript accordingly. We sincerely appreciate the reviewers' insightful and constructive feedback, which has significantly improved the clarity and technical rigor of our work.*

---

### Author Response · Authors · 2025-12-03
**Positive Points from Reviewers**

*Dear AC and Reviewers, to assist your evaluation of our submission, we provide below a consolidated summary of the positive assessments highlighted across all reviews.*

**Importance & Relevance of the Problem**
- Addresses a *highly challenging and relevant* problem in cooperative MARL with sparse episodic rewards. **(Reviewer C2ir, cDaS)**
- Sparse-reward credit assignment is a *significant and important research direction*. **(Reviewer cDaS)**

**Novelty & Core Idea**

- Unified framework combining **online inverse preference learning**, **value decomposition**, and **CTDE PPO-style training** is **novel and interesting**. **(Reviewer C2ir, aXhn, cDaS)**

- Transforming sparse returns into **trajectory preferences**, then learning implicit dense rewards via IPL in Q-space, is **creative and novel**. **(Reviewer C2ir, e5ym, cDaS)**

- Integrating ideas from PbRL + MARL is a **novel cross-field contribution**. **(Reviewer cDaS)**

**Dual-Advantage Architecture**

- Introduction of **dual advantage streams** (global critic advantage + local actor advantages) is a **key contribution** for improving multi-agent credit assignment. **(Reviewer C2ir)**

- Local advantages aligned with global objectives address a known weakness of MAPPO. **(Reviewer aXhn)**

- The design is called **elegant**, **interesting**, and **promising** for credit assignment. **(Reviewer cDaS)**

**Theoretical Contributions**

- Proposition 4.1 and Theorem 4.2 show that the learned implicit reward converges to a form **behaviorally indistinguishable from the true reward**, preserving optimal policies. **(Reviewer C2ir, aXhn)**

- Reviewers recognize the theory as providing **policy alignment** and **sound reward recovery guarantees**. **(Reviewer e5ym)**

**Empirical Performance**

- Empirical results are described as **very strong**. **(Reviewer C2ir)**

- IMAP-Rule and IMAP-LLM **significantly outperform all baselines** in the experiments. **(Reviewer C2ir)**

- Strong win-rate/return and sample efficiency gains reported on **SMACv2** and **MAMuJoCo**. **(Reviewer aXhn, e5ym)**

- Evaluations cover both **discrete** and **continuous** MARL tasks. **(Reviewer e5ym)**

**LLM-based Preference Feedback**

- The exploration of LLM-generated preferences is **timely**, **promising**, and **valuable** for settings without explicit rewards. **(Reviewer e5ym, cDaS)**

- LLM-based preferences provide **denser and more nuanced feedback**, improving scalability. **(Reviewer e5ym)**

**Quality of Writing & Clarity**

- The formulation and writing are described as **clear and easy to follow**. **(Reviewer aXhn)**

- The motivation for applying preference learning to sparse MARL is **sound**. **(Reviewer e5ym)**

**Completeness of Experiments**

- Experiments span discrete + continuous tasks with **reasonable baselines**, **ablations**, and **rule vs. LLM comparisons**. **(Reviewer e5ym)**

- Demonstrates potential for LLM-based preference learning in real-world tasks where dense rewards are not available. **(Reviewer e5ym)**

***

In summary, across reviewers, we see that the paper is recognized for introducing a novel and impactful solution to sparse-reward cooperative MARL. The transformation from episodic returns to trajectory preferences, combined with an implicit reward model and dual advantage streams, is consistently praised as creative, elegant, and effective. Theoretical guarantees reinforce the method’s soundness, while strong empirical results demonstrate effectiveness in both discrete and continuous domains. Reviewers particularly value the exploration of LLM-based preference signals as a practical and forward-looking extension.

*We would like to note that, although Reviewers C2ir and cDaS recommended rejection, we appreciate that both acknowledged the **significance** of the problem we address and **the novelty of our proposed framework**. Importantly, their reviews contain several positive assessments regarding the contribution, including the relevance of tackling sparse-reward cooperative MARL, the creativity of integrating inverse preference learning with value decomposition, and the promise of our dual-advantage design.*

We also note that the main reasons they raised for rejection stem largely from issues of **clarity, justification, or missing baselines**. In our rebuttal, we provide specific and comprehensive responses and updates to paper that directly address each of these concerns. We believe the clarifications, additional explanations, and corrections we provide adequately resolve the issues that motivated their recommendations.

---

### Meta-Review · Area_Chair_qwne · 2026-01-10

**Summary:**

The paper proposes IMAP, a framework integrating online Inverse Preference Learning (IPL) with MARL to address sparse reward challenges. Despite the interesting motivation of combining PbRL and MARL, the submission is recommended for Rejection. This decision is based on flaws in the architectural design (specifically the redundancy of the Dual-Critic system), the use of restrictive function approximators (linear mixing), and the lack of rigorous empirical validation against state-of-the-art sparse-reward baselines on the primary benchmarks. While the authors tried to address these issues in the rebuttal, the core concerns regarding efficiency, scalability, and empirical superiority remain unresolved.

**Reviewer Concerns:**

**Concerns addressed by rebuttal:**
- Additional comparisons with HAPPO were provided (Reviewer e5ym).
- The discrepancy in Theorem statement was fixed, and the distinction between temperature parameters was clarified (Reviewer cDaS).
- LLM implementation details: Sensitivity analysis for LLM prompts was added, and clarifications on local execution/costs were provided (Reviewers C2ir, aXhn).
- Explanations regarding the sparse reward setup in MAMuJoCo and the definition of local rewards were updated (Reviewers e5ym, C2ir).


**Partially addressed or outstanding concerns**
- Reviewers cDaS, argued that the main results on SMACv2 are unsubstantiated without comparison to relevant sparse-reward MARL baselines like STAS, AREL, and TAR2. The authors added these comparisons only in the simplified SMACLite environment (Appendix C.4), claiming these baselines are not scalable to SMACv2 or MAMuJoCo. This defense is weak: it effectively admits that the proposed method's superiority is not demonstrated against SOTA on the paper's main testbeds. The claim of being a new SOTA framework remains unproven on the challenging tasks presented in the main text. Given that the referred baselines use neural function approximations, there is no fundamental limit to high-dimensional application. Future revisions should either technically justify these constraints or include the comparisons to substantiate SOTA claims.

- Architectural inefficiency & redundancy (Dual-Critic) (Reviewer cDaS): The reviewer cDaS criticized the Dual-Critic design: learning a value function for reward recovery $V_{reward}$ and then training a separate centralized critic $V_{policy}$ for PPO, as redundant, inefficient, and a sign of a "bolted-on" implementation. The authors defended this by stating that preference-learned values are inaccurate, necessitating a separate policy critic. This acknowledgement validates the reviewer's concern that the preference learning module is inefficient and not well-integrated enough to guide the policy directly. The structural redundancy and computational overhead remain negatives.

- Restrictive linear mixer assumption (Reviewers C2ir, cDaS): Reviewers questioned the dependence on a simple linear mixing network, noting it limits expressiveness compared to standard non-linear mixers used in complex MARL tasks. The authors argued that linear mixers prevent overfitting and preserve additive decomposition. However, this is a heuristic defense: the authors should have provided empirical ablation studies comparing their linear approach against standard non-linear mixers to substantiate the claim that restricted expressiveness does not hurt performance in this context. Without such direct comparisons or a theoretical extension to monotonic functions, the concern that the framework sacrifices representational power for theoretical convenience remains.

- Practicality of theoretical guarantees (Reviewers C2ir, aXhn): The theoretical proofs (Theorem 1) rely on asymptotic assumptions (infinite samples, full coverage of trajectory pairs) that are impossible to meet in practice. The authors acknowledged these are asymptotic results. While common in theory papers, the gap between these idealized assumptions and the practical finite-sample performance of the algorithm remains a valid concern for the reviewers.

**Reviewer Scores:**

- Reviewer C2ir (4): Likely to remain 4. The rebuttal defended the linear mixer choice but did not empirically prove its sufficiency against non-linear baselines in complex scenarios.
- Reviewer e5ym (6): Likely to remain 6. The authors addressed the specific request for HAPPO comparisons and clarified environment settings.
- Reviewer aXhn (6): Likely to remain 6. The theoretical concerns were acknowledged as asymptotic limitations, likely maintaining the reviewer's marginal support.
- Reviewer cDaS (2): Likely to remain 2. The reviewer raised fundamental flaws in the architecture ("bolted on" design) and experimental rigor. While factual errors were fixed, the core argument that the method is inefficient and lacks sufficient benchmarking on complex tasks remains valid.

---

### Decision · Program_Chairs · 2026-01-26

Reject